# REGRESSION-BASED TEST-TIME ADAPTATION OF VISION-LANGUAGE MODELS

## ABSTRACT

Mainstream Test-Time Adaptation (TTA) techniques aim to select confident views with lower entropy from a set of augmented views to process instance-level adaptation for vision-language models, *e,g*, CLIP. However, entropy-based strategy, relying only on the current instance probability distribution, struggles to estimate reliable entropy for outliers. Surprisingly, we observe that using ground-truth cross-entropy loss on labeled data to select confident views can achieve overwhelming performance, which motivates us to directly establish a regression mapping between augmented views and their corresponding cross-entropy loss. This paper proposes **R**egression-based **T**est-time **A**daptation (**RTA**) that exploits such view-loss relationships as a 'free lunch' for CLIP-based image classification. By training a regression model on diversely distributed pseudo-labeled data independent of downstream tasks, we can predict the pseudo cross-entropy loss for each augmented view during actual TTA, thereby achieving more accurate view selection without access to true labels. The significant advantage of RTA is that the view-loss mapping relationship can be estimated in advance on diverse data, avoiding the current methods that rely solely on the probability distribution of a single test instance. Extensive experiments on multiple single-label, multi-label, and cross-domain benchmarks show that RTA significantly outperforms existing entropy-based TTA methods with negligible additional cost. Our code is available at https://anonymous.4open.science/r/RTA-2ADD.

## 1 INTRODUCTION

Vision-language models (VLMs) (Radford et al., 2021; Zeng et al., 2024) have demonstrated unprecedented success in bridging visual and textual modalities on massive web-scale datasets (Sharma et al., 2018; Schuhmann et al., 2022). Traditional prompt learning works (Zhou et al., 2022; Hu et al., 2023; Xing et al., 2024) require extensive labeled data and computational resources, making them impractical for many scenarios. Beyond that, Test-Time Adaptation (TTA) (Shu et al., 2022; Feng et al., 2023; Zhu et al., 2024b; Farina et al., 2024; Zhao et al., 2024; Zhou et al., 2025; Wu et al., 2025) has emerged as a compelling paradigm that enables models to adapt to new domains using only unlabeled test data without accessing source model parameters and training data.

Generally, the task of TTA typically involves accurately selecting confident views from a set of augmented views, aiming to reduce the inconsistency and uncertainty of the model. As depicted in Figure 1(a), current sophisticated algorithms, denoted as $f$, such as entropy minimization (Shu et al., 2022; Farina et al., 2024), image diffusion (Feng et al., 2023), CLIP reward (Zhao et al., 2024), and bound entropy minimization (Wu et al., 2025), are predominantly designed to implement view selection strategies. Subsequently, the final predicted result is integrated from all logits of these selected views. To obtain reliable views, these methods strive to excavate discriminative and distinguishing class information from limited individual instances during TTA.

The core idea of our method is illustrated in Figure 1(b). We discover a significant regression mapping relationship between the logits of augmented views and the label cross-entropy loss. Specifically, the smaller the label cross-entropy loss, the more accurate the predictions brought by the corresponding views, regardless of the distribution of test instances. Our research findings offer a new direction for the study of TTA algorithms: the information sources that determine reliable views are not limited to the current single test instance.

We propose a novel **R**egression-based **T**est-time **A**daptation (**RTA**) method for vision-language models, aiming to investigate how regression mapping trends can enhance TTA performance. The core insight of RTA is to leverage regression mapping as a "free lunch": it directly computes regression loss from the logits of augmented views, with views yielding smaller regression loss selected as confident views. Essentially, RTA establishes a regression mapping relationship across unlabeled data with maximally broad and diverse distributions. This design is conducive to strengthening the recognition capability of test-time adaptation frameworks.

A key limitation of existing TTA algorithms lies in their reliance on information from individual test instances, regardless of their architectural complexity. Although some excellent works have designed memory modules and cache mechanisms to provide additional information guidance, these methods need to continuously update and maintain dynamically changing historical samples. Once the distribution of test instances deviates significantly from the historical distribution, these methods will immediately fail. Crucially, the regression

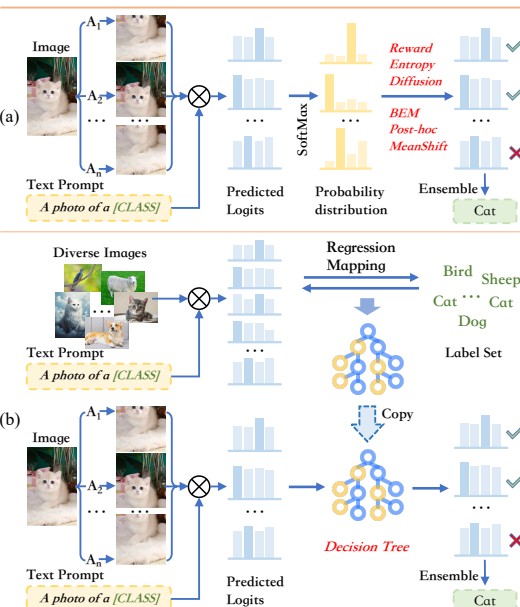

Figure 1: (a). Current entropy-based TTA methods often design complex modules to accurately select confident views. (b). RTA treats the regression mapping as a "free lunch" and directly predicts the regression loss from the logits of the views, with the views with smaller regression loss being selected as confident views.

mapping of RTA only needs to be trained once in the initial stage, and then it can directly adapt to test instances with arbitrary distributions, making it more practical in real-world applications. In summary, the contributions of this work are threefold:

- **Key Finding**: We discover a strong regression mapping between optimal augmented views and label cross-entropy loss, *i.e.*, lower loss consistently indicates more accurate predictions across all distributions.

- **Method**: The proposed **R**egression-based **T**est-Time **A**daptation (RTA) leverages this regression relationship to select confident views directly, eliminating complex algorithmic designs required by existing TTA methods.

- **Advantage**: Unlike current methods, limited to single test instances, RTA trains once on diverse unlabeled data and adapts to any test distribution without updates.

## 2 RELATED WORKS

**Test-Time Adaptation.** Test-time adaptation (TTA) directly addresses the critical challenge of distribution shift encountered during model deployment. Its core objective is to enhance the robustness using only unlabeled data from the test stream (Zhang et al., 2022; Shu et al., 2022; Karmanov et al., 2024; Zhao et al., 2024; Zhu et al., 2024b; Zhou et al., 2025; Wu et al., 2025). The field has evolved from basic batch normalization layer parameter update methods such as TENT (Wang et al., 2021) to more sophisticated methods, including customized instance-level tuning (Shu et al., 2022), diffusion generation of pseudo images (Feng et al., 2023), streaming adaptation for temporal scenes (Lee & Chang, 2024; Liu et al., 2024), designing positive/negative cache to assist subsequent adaptation (Karmanov et al., 2024), and a theoretical framework for analyzing the impact of entropy minimization on prediction confidence (Farina et al., 2024). Recent studies have extended TTA to handle complex scenarios such as multi-label classification (Wu et al., 2025), incorporated Bayesian uncertainty estimation (Zhou et al., 2025), dynamically sampled the optimal prompt from prompt sets (Xiao et al., 2025), and developed debias strategies (Song et al., 2023; Zhu et al., 2024a). These methods mainly rely on self-supervision objectives to encourage confident predictions while maintaining model stability across different test distributions.

**Regression in Deep Learning.** Regression aims at modeling continuous target variables. Classical approaches include linear regression (Schleich et al., 2016), support vector regression (Anand et al., 2020), and decision tree methods (Fong & Motani, 2024). Recent studies have applied regression to scenarios such as transfer learning and multimodal uncertainty estimation (Maddox et al., 2021; Ruppel et al., 2023; Parente et al., 2024; Zeng, 2024), but most rely on domain-specific optimizations or prior knowledge of the target distribution. The most closely related work to RTA is the loss predictor for test-time augmentation proposed by Kim et al (Kim et al., 2020), which is trained in a supervised image classification setting. They employ a deep neural network loss predictor for single-view augmentation selection, focusing mainly on optimizing test-time image augmentation strategies, and require training within the target domain using target-domain samples. In contrast, RTA aims to adapt CLIP to arbitrary target domains by selecting high-confidence views without ground-truth labels. We use a lightweight tree-based regression model that only needs to be trained once offline, and can then be applied directly to any target domain.

## 3 PRELIMINARIES

Consider a training set $\mathcal{D}^{\text{train}} = \{(\mathbf{x}^{\text{train}}, \mathbf{y}^{\text{train}}) \mid \mathbf{x}^{\text{train}} \in \mathcal{X}^{\text{train}}, \mathbf{y}^{\text{train}} \in \mathcal{Y}^{\text{train}}\}^{M^{\text{train}}}$ and a downstream test set $\mathcal{D}^{\text{test}} = \{(\mathbf{x}^{\text{test}}, \mathbf{y}^{\text{test}}) \mid \mathbf{x}^{\text{test}} \in \mathcal{X}^{\text{test}}, \mathbf{y}^{\text{test}} \in \mathcal{Y}^{\text{test}}\}^{M^{\text{test}}}$ with arbitrary distributions deviating from training set. The standard Test-Time Adaptation (TTA) process mainly consists of *random view augmentation*, *confident view selection*, and *confident view integration*. Mainstream TTA methods focus on the second stage, *i.e.*, how to obtain accurate and reliable views.

**Random view augmentation**. Given a test instance $\mathbf{x}^{\text{test}}$ from $\mathcal{D}^{\text{test}}$ and a set $\mathcal{A}$ of $N$ random augmentation functions, $\mathbf{x}^{\text{test}}$ is first augmented $N$ times to generate a set of different augmented views, denoted as $\mathbf{X}^{\text{test}} = \{\mathbf{x}_i^{\text{test}} \mid \mathbf{x}_i^{\text{test}} = \mathcal{A}_i(\mathbf{x}^{\text{test}})\}_{i=1}^N$.

**Confident views selection**. Entropy-based methods, which directly select views with low entropy as confident views, aim to enhance the certainty of model predictions. The Shannon Entropy (SE) of each augmented view is defined as:

$$\mathbf{H}_{\text{SE}}(\mathbf{P}(\cdot|\mathbf{x}_i^{\text{test}})) = -\sum_{l=1}^{L} \mathbf{P}(y = l|\mathbf{x}_i^{\text{test}}) \log[\mathbf{P}(y = l|\mathbf{x}_i^{\text{test}})], \qquad (1)$$

where $l \in \mathcal{Y}^{\text{test}}$ represents class labels, and $L$ is the number of labels in the test set. TPT (Shu et al., 2022), DiffTPT (Feng et al., 2023), and RLCF (Zhao et al., 2024) update specific prompts for each sample using the entropy loss of confident views. Zero (Farina et al., 2024) provides a solid theoretical derivation for the effectiveness of entropy minimization while eliminating the cumbersome steps of parameter updates. However, relying solely on the current single instance to determine confident views is obviously limited in information.

**Confident views ensemble.** After parameter updates, TPT, DiffTPT, RLCF, and ML-TTA will re-input the current test instance and obtain prediction results. In contrast, Zero derives the prediction result by directly integrating confident views. Once the inference for one instance is completed, the model will immediately adapt to subsequent test instances. Entropy minimization has become the *de facto* standard uncertainty measure for confident view selection in modern TTA.

## 4 METHODS

Our method consists of two stages. (1). Explore a reference unlabeled database with distribution diversity and learn a regression mapping function that can correlate views with pseudo-label cross-entropy loss. (2). Standard Test-Time Augmentation process, with the difference being that the optimal confidence view is selected through the learned function.

### 4.1 DISCUSSION OF THE REGRESSION RELATIONSHIP

Our intuition is this: Since entropy loss can serve as a benchmark for selecting confident views in unlabeled scenarios, using actual label cross-entropy loss to select views in labeled scenarios will undoubtedly outperform entropy loss by a significant margin — this is because the former can directly eliminate interference from other incorrect labels.

Table 1: Performance improvement of `SE` and `LCE` for `RN50` with different numbers of views.

|  | IN-1k | IN-A | IN-V | IN-R | IN-K |
|---|---|---|---|---|---|
| RN50 | 59.8 | 23.2 | 52.9 | 60.7 | 35.5 |
| $\mathbf{H}_{\mathrm{SE}}(8)$ | 60.3 | 31.5 | 53.8 | 61.0 | 35.6 |
| $\mathbf{H}_{\mathrm{LCE}}(8)_\uparrow$ | $75.4_{15.1}$ | $50.2_{18.7}$ | $69.5_{15.7}$ | $73.7_{12.7}$ | $49.6_{14.0}$ |
| $\mathbf{H}_{\mathrm{SE}}(16)$ | 60.7 | 33.1 | 54.1 | 61.4 | 35.8 |
| $\mathbf{H}_{\mathrm{LCE}}(16)_\uparrow$ | $78.5_{17.8}$ | $58.7_{25.6}$ | $74.0_{19.9}$ | $77.9_{16.5}$ | $53.9_{18.1}$ |
| $\mathbf{H}_{\mathrm{SE}}(32)$ | 61.7 | 34.7 | 55.5 | 61.5 | 37.1 |
| $\mathbf{H}_{\mathrm{LCE}}(32)_\uparrow$ | $82.4_{20.7}$ | $65.1_{30.4}$ | $77.6_{22.1}$ | $81.1_{19.6}$ | $57.7_{20.6}$ |
| $\mathbf{H}_{\mathrm{SE}}(64)$ | 61.9 | 35.7 | 55.8 | 61.8 | 38.4 |
| $\mathbf{H}_{\mathrm{LCE}}(64)_\uparrow$ | $85.8_{23.9}$ | $70.9_{35.2}$ | $80.4_{24.6}$ | $83.7_{21.9}$ | $62.3_{23.9}$ |

Table 2: Performance improvement of `SE` and `LCE` for `ViT-B/16` with different numbers of views.

|  | IN-1k | IN-A | IN-V | IN-R | IN-K |
|---|---|---|---|---|---|
| ViT-B/16 | 66.7 | 47.8 | 60.8 | 73.9 | 46.1 |
| $\mathbf{H}_{\mathrm{SE}}(8)$ | 69.4 | 60.1 | 63.6 | 78.8 | 48.6 |
| $\mathbf{H}_{\mathrm{LCE}}(8)_\uparrow$ | $81.6_{12.2}$ | $76.8_{16.7}$ | $77.2_{13.6}$ | $88.5_{9.7}$ | $62.3_{13.7}$ |
| $\mathbf{H}_{\mathrm{SE}}(16)$ | 69.6 | 62.4 | 63.4 | 79.1 | 48.8 |
| $\mathbf{H}_{\mathrm{LCE}}(16)_\uparrow$ | $84.9_{15.3}$ | $83.0_{20.6}$ | $80.6_{17.2}$ | $91.1_{12.0}$ | $66.8_{18.0}$ |
| $\mathbf{H}_{\mathrm{SE}}(32)$ | 70.3 | 63.6 | 64.5 | 80.2 | 49.8 |
| $\mathbf{H}_{\mathrm{LCE}}(32)_\uparrow$ | $87.6_{17.3}$ | $87.1_{23.5}$ | $83.8_{19.3}$ | $92.9_{12.7}$ | $70.3_{20.5}$ |
| $\mathbf{H}_{\mathrm{SE}}(64)$ | 70.6 | 64.3 | 65.2 | 80.4 | 50.1 |
| $\mathbf{H}_{\mathrm{LCE}}(64)_\uparrow$ | $89.6_{19.0}$ | $90.2_{25.9}$ | $86.6_{21.4}$ | $94.4_{14.0}$ | $73.4_{23.3}$ |

**Ceiling TTA**. Therefore, we first explore the maximum performance improvement of TTA using actual label cross-entropy loss on multiple datasets with true labels, which we refer to as "Ceiling TTA". Given a set of augmented views of a test instance $\mathbf{X}^{\mathrm{test}} = \{\mathbf{x}_i^{\mathrm{test}}\}_{i=1}^N$ and its corresponding label $\mathbf{y}^{\mathrm{test}} \in \mathcal{Y}^{\mathrm{test}}$, similar to the entropy loss shown in Equation 1, the actual label cross-entropy loss (LCE) can be expressed more concisely as the negative log-probability of the true label:

$$\mathbf{H}_{\mathrm{LCE}}(\mathbf{y}^{\mathrm{test}}, \mathbf{P}(\cdot|\mathbf{x}_i^{\mathrm{test}})) = -\log[\mathbf{P}(y = \mathbf{y}^{\mathrm{test}}|\mathbf{x}_i^{\mathrm{test}})]. \tag{2}$$

This loss directly measures the consistency between the model's prediction and the ground-truth label. Lower loss values indicate more accurate and reliable views. The final prediction is obtained by ensembling the top-$k$ views with the lowest label cross-entropy loss. As shown in Tables 1 and 2, $\mathbf{H}_{\mathrm{LCE}}$ achieves significant performance gains across all datasets. For instance, when using CLIP-ViT-B/16 with 64 augmented views, the accuracy on ImageNet-A/ImageNet-R reaches 90.2%/94.4%. Confident view selection with LCE achieves very high accuracy that is near-saturation with respect to the number of augmented views. This observation motivates us to explore the direct correspondence between each view and its label cross-entropy loss, as such a mapping can guide confident view selection and potentially benefit other test-time adaptation frameworks.

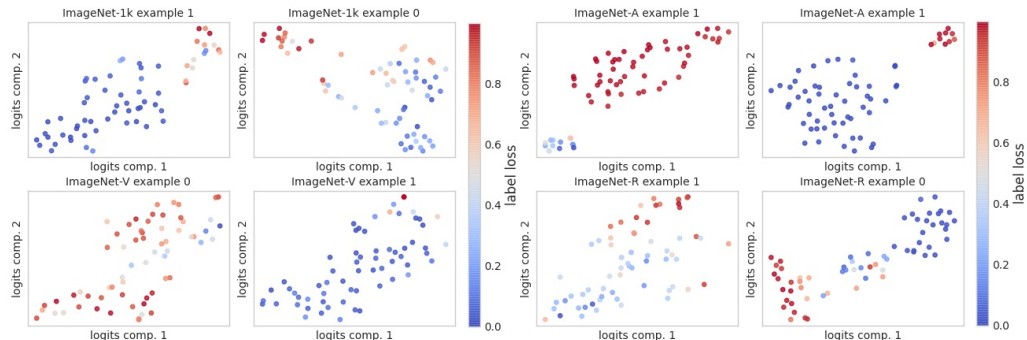

Figure 2: T-SNE of all instance views' logits reveals their 2D distribution has significant (likely non-linear) structural correlations to normalized label cross-entropy loss, can be well-fitted by regression models.

**Logits-loss Visualization.** We perform t-SNE visualization on all views of individual instances along with their corresponding losses (as shown in Figure 2). Each dot in the figure represents the 2D coordinates of a view's logits after dimensionality reduction, while the shade of color corresponds to the normalized loss value of that view. From the figure, in the distribution plots of all datasets, there are obvious color clustering or gradient phenomena, indicating that there exists a significant structural relationship between logits and label cross-entropy loss (and this relationship is most likely non-linear), and regression models are highly suitable at fitting such relationships.

**Spearman's Rank Correlation**. We analyzed the top 10 features (i.e., the top logits with the highest correlation to the label cross-entropy loss) using Spearman's Rank Correlation, as shown in Figure 3. The figure reports the correlation coefficients and p-values for each feature, indicating the strength and statistical significance of their monotonic relationships with the labels. The coefficients range from near $-1$ to 1, revealing strong positive, strong negative, or weak/no correlations, while the p-values assess their significance. This indicates that there are monotonic relationships of different degrees between these features and the labels.

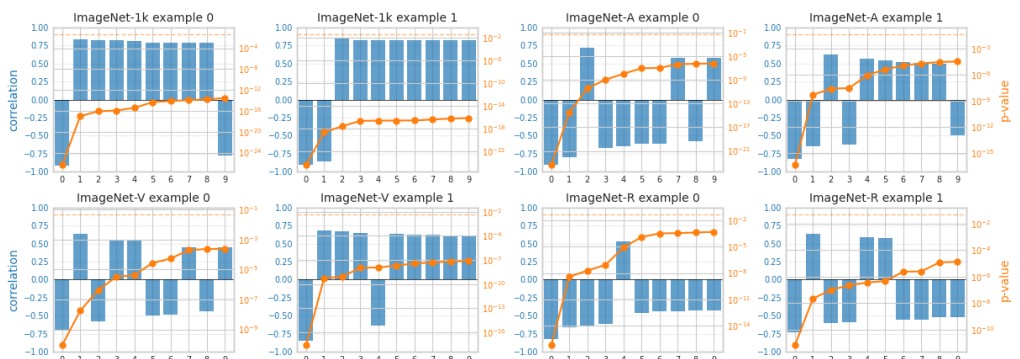

Figure 3: Spearman's analysis of top 10 features vs labels shows varying correlation coefficients ($-1$ to $1$) and p-values, indicating monotonic relationships with distinct statistical significance.

## 4.2 REGRESSION MAPPING LEARNING

In this section, we will elaborate on how to learn the regression mapping relationship between views and their corresponding label cross-entropy losses. Note that although we explore the regression correlation on the augmented views of the original image in the previous section, the original image itself can actually be regarded as a view. Therefore, we only need to learn the regression mapping function based on the original image and the pseudo-label cross-entropy loss (we obtain pseudo-labels by filtering high-confidence samples to avoid accessing the ground-truth), without the need for additional data augmentation, which greatly reduces the computational cost of fitting the regression function.

Consider a regression set $\mathcal{D}^{\mathrm{reg}} = \{(\mathbf{x}^{\mathrm{reg}}, \mathbf{y}^{\mathrm{reg}}) \mid \mathbf{x}^{\mathrm{reg}} \in \mathcal{X}^{\mathrm{reg}}, \mathbf{y}^{\mathrm{reg}} \in \mathcal{Y}^{\mathrm{reg}}\}^{M^{\mathrm{reg}}}$. For a regression sample $(\mathbf{x}^{\mathrm{reg}}, \mathbf{y}^{\mathrm{reg}})$, we initialize the prompts with template "*a photo of* $[\mathbf{CLS}]_{\mathbf{j}}$", in which $[\mathbf{CLS}]_{\mathbf{j}}$ represents the $j$-th label name, e.g., dog or cat, results in $\mathbf{t}_j$, the logit for $j$-th class of $\mathbf{x}^{\mathrm{reg}}$ is computed as:

$$s_j^{\mathbf{x}^{\mathrm{reg}}} = \langle \mathrm{Enc}^{\mathrm{I}}(\mathbf{x}^{\mathrm{reg}}), \mathrm{Enc}^{\mathrm{T}}(\mathbf{t}_j) \rangle, \tag{3}$$

where $\mathrm{Enc}^{\mathrm{I}}$ and $\mathrm{Enc}^{\mathrm{T}}$ represent the frozen image encoder and text encoder of CLIP, $\langle \cdot, \cdot \rangle$ signifies the dot product. The pseudo-label is $\mathbf{y}^{\mathrm{reg}}$, which corresponds to the class index $l$, the pseudo-label cross-entropy loss is given by the negative log-probability of the $\mathbf{y}^{\mathrm{reg}}$:

$$\mathcal{L}_{\mathrm{CE}}(\mathbf{y}^{\mathrm{reg}}|s^{\mathbf{x}^{\mathrm{reg}}}) = -\log\left(\frac{\exp(s_l^{\mathbf{x}^{\mathrm{test}}})}{\sum_{k=1}^{L}\exp(s_k^{\mathbf{x}^{\mathrm{reg}}})}\right). \tag{4}$$

In this paper, we adopt the classic regression decision tree algorithm to fit the input features $s^{\mathbf{x}^{\mathrm{reg}}}$ and the target values $\mathcal{L}_{\mathrm{CE}}(\mathbf{y}^{\mathrm{reg}}|s^{\mathbf{x}^{\mathrm{reg}}})$. The fitting function is defined as:

$$f(s^{\mathbf{x}^{\mathrm{reg}}}) = \sum_{m=1}^{M} c_m \cdot \mathbb{I}(s^{\mathbf{x}^{\mathrm{reg}}} \in R_m), \tag{5}$$

where $M$ is the number of leaf nodes, $R_m$ is the input space region corresponding to the $m$-th leaf node, $c_m$ is the predicted value of the $m$-th leaf node, and $\mathbb{I}(\cdot)$ is the indicator function. The predicted value $c_m$ of the $m$-th leaf node is defined as the average of $\mathcal{L}_{\mathrm{CE}}(\mathbf{y}^{\mathrm{reg}}, s^{\mathbf{x}^{\mathrm{reg}}})$ of all training samples that fall into the region $R_m$ of this leaf node:

$$c_m = \frac{1}{|R_m|} \sum_{w:s^{\mathbf{x}^{\mathrm{reg}}} \in R_m} \mathcal{L}_{\mathrm{CE}}^{w}(\mathbf{y}^{\mathrm{reg}}|s^{\mathbf{x}^{\mathrm{reg}}}). \tag{6}$$

We use the mean squared error to determine the optimal predicted value for each leaf. Then, the optimization objective of the entire tree is to minimize the mean squared error between the pseudo cross-entropy losses of all training samples and the cross-entropy losses predicted by the decision tree:

$$\sum_{m=1}^{M} \sum_{w:s^{\mathbf{x}^{\mathrm{reg}}} \in R_m} \left[\mathcal{L}_{\mathrm{CE}}^{w}(\mathbf{y}^{\mathrm{reg}}|s^{\mathbf{x}^{\mathrm{reg}}}) - c_m\right]^2. \tag{7}$$

---

**Algorithm 1** Regression Mapping via Decision Tree

---

1: **Input:** Regression set $\mathcal{D}^{\text{reg}} = \{(\mathbf{x}^{\text{reg}}, \mathbf{y}^{\text{reg}}) \mid \mathbf{x}^{\text{reg}} \in \mathcal{X}^{\text{reg}}, \mathbf{y}^{\text{reg}} \in \mathcal{Y}^{\text{reg}}\}^{M^{\text{reg}}}$, encoder $\text{Enc}^{\text{I}}/\text{Enc}^{\text{T}}$.
2: **Output:** A trained regression decision tree $f$.
3: Text prompt $\mathbf{t}_j \leftarrow$ "*a photo of* $[\mathbf{CLS}]_j$" for each class. Initialize empty regression set $\mathcal{D}^{\text{tree}}$.
4: **for** each sample $(\mathbf{x}^{\text{reg}}, \mathbf{y}^{\text{reg}})$ in $\mathcal{D}^{\text{reg}}$ **do**
5:     Let $l$ be the class index corresponding to $\mathbf{y}^{\text{reg}}$.
6:     **for** $j = 1, \dots, L$ **do**
7:         Compute $s_j^{\mathbf{x}^{\text{reg}}}$ for $j$-th class using Eq.(3).
8:     **end for**
9:     Compute $\mathcal{L}_{\text{CE}}(\mathbf{y}^{\text{reg}} | s^{\mathbf{x}^{\text{reg}}})$ using Eq.(4).
10:    Add the pair $(s^{\mathbf{x}^{\text{reg}}}, \mathcal{L}_{\text{CE}})$ to $\mathcal{D}^{\text{tree}}$.
11: **end for**
12: Train a regression decision tree $f$ using fitting function 5 on $\mathcal{D}^{\text{tree}}$. The tree is fitted to minimize the Mean Squared Error using Eq.(7).
13: **return** $f$

---

**Algorithm 2** Regression-based Test-time Adaptation (RTA)

---

1: **Input:** Test instance $\mathbf{x}^{\text{test}}$, a set of $N$ augmentation functions $\mathcal{A} = \{\mathcal{A}_i\}_{i=1}^N$, image/text encoder $\text{Enc}^{\text{I}}/\text{Enc}^{\text{T}}$, the trained regression decision tree $f$.
2: **Output:** The final prediction for $\mathbf{x}^{\text{test}}$.
3: Generate augmented views $\mathbf{X}^{\text{test}} \leftarrow \{\mathcal{A}_i(\mathbf{x}^{\text{test}})\}_{i=1}^N$.
4: Initialize the temporary set $\mathcal{D}^{\text{pred}}$.
5: **for** each augmented view $\mathbf{x}_i^{\text{test}}$ in $\mathbf{X}^{\text{test}}$ **do**
6:     **for** $j = 1, \dots, L$ **do**
7:         Compute $s_{ij}^{\mathbf{x}^{\text{test}}}$ for $j$-th class using Eq.(8).
8:     **end for**
9:     Predict the cross-entropy loss for the view using the decision tree: $\mathcal{L}_i^{\text{pred}} \leftarrow f(\mathbf{s}^{\mathbf{x}_i^{\text{test}}})$.
10:    Add the pair $(\mathbf{x}_i^{\text{test}}, \mathcal{L}_i^{\text{pred}})$ to a temporary set $\mathcal{D}^{\text{pred}}$.
11: **end for**
12: Select the top-$k$ views in $\mathcal{D}^{\text{pred}}$ with the smallest predicted losses to get $V_{\text{conf}}$ using Eq.(10).
13: **return** Average the predictions in $V_{\text{conf}}$.

---

## 4.3 REGRESSION-BASED TEST-TIME ADAPTATION (RTA)

Once we obtain the regression decision tree, we can directly predict the label cross-entropy loss of each augmented view and select the optimal confidence view in the second stage.

Like other TTA methods, RTA first obtains a set of augmented views $\mathbf{X}^{\text{test}} = \{\mathbf{x}_i^{\text{test}} \mid \mathbf{x}_i^{\text{test}} = \mathcal{A}_i(\mathbf{x}^{\text{test}})\}_{i=1}^N$ of the test instance $x^{\text{test}}$, and then uses the same vision-language model as in the first stage to get the logit for each class of each view and $l$ text prompts:

$$s_{ij}^{\mathbf{x}_i^{\text{reg}}} = \langle \text{Enc}^{\text{I}}(\mathbf{x}_i^{\text{reg}}), \text{Enc}^{\text{T}}(\mathbf{t}_j) \rangle, \tag{8}$$

Then, substituting $s_i^{\mathbf{x}_i^{\text{reg}}}$ into Equation 5 allows us to obtain the predicted label cross-entropy loss for each view:

$$f(s_i^{\mathbf{x}^{\text{reg}}}) = \sum_{m=1}^M c_m \cdot \mathbb{I}(s_i^{\mathbf{x}^{\text{reg}}} \in R_m) \tag{9}$$

The top-$k$ views with the smallest losses form the optimal confidence view set:

$$V_{\text{conf}} = \left\{ \mathbf{x}_i^{\text{reg}} \mid i \in \arg \min_{i=1,2,\dots,N} \left\{ f(s_i^{\mathbf{x}^{\text{reg}}}) \right\}_{\text{top } k} \right\} \tag{10}$$

The final prediction result is obtained by directly fusing the views in $V_{\text{conf}}$. We summarize the two stages of our method through Algorithms 1 and 2.

## 5 EXPERIMENTS

### 5.1 EXPERIMENTAL SETUP

**Benchmarks.** RTA aims to learn the mapping function from logits to label cross-entropy loss, which is independent of any downstream classification task. Therefore, we include two benchmarks: single-label following Zero (Farina et al., 2024), and multi-label following ML-TTA (Wu et al., 2025). The single-label datasets include ImageNet (Deng et al., 2009), ImageNet-A (Hendrycks et al., 2019), ImageNet-V2 (Recht et al., 2019), ImageNet-R (Hendrycks et al., 2020), ImageNet-K (Wang et al., 2019) and 10 cross domain datasets (*e.g.*, Cars (Krause et al., 2013), Pets (Parkhi et al., 2012), and Aircraft (Maji et al., 2013)). The multi-label datasets include MSCOCO (Lin et al., 2014), VOC2007 (Everingham et al., 2010), and NUSWIDE (Chua et al., 2009).

**Implementation Details** We use the pre-trained CLIP model with `RN50` and `ViT-B/16` as the backbone. For the first stage, we selecte ImageVal-12k as the regression mapping data. To balance performance and efficiency, we only sample $1,000$ examples (sampling by logit-based equal-interval from $5,000$ samples with threshold $\geq 0.8$ on CLIP's predicted confidence) for training, although more data can yield higher performance. We use LightGBM (Ke et al., 2017) as the regression model, training to convergence after 100 rounds. The maximum depth is 5, the maximum number of leaves is 16, the learning rate is 0.01, and all other parameters are set to default. For the second stage, the TTA process follows the settings of Zero and ML-TTA, with the number of augmented views $N = 64$ and the confidence-based filtering ratio 0.1. All experiments are conducted on a single NVIDIA V100 GPU.

Table 3: Comparison of the accuracy (%) of CLIP `RN50` and `ViT-B/16` on ImageNet and its variants with the state-of-the-art methods.

| | Method | IN-1k | IN-A | IN-V2 | IN-R | IN-S | Average | OOD Avg |
|---|---|---|---|---|---|---|---|---|
| | CLIP [ICML 2022] | 59.81 | 23.24 | 52.91 | 60.72 | 35.48 | 46.43 | 43.09 |
| RN50 | TPT [NeurIPS 2022] | 60.74 | 26.67 | 54.70 | 59.11 | 35.09 | 47.26 | 43.89 |
| | DiffTPT [ICCV 2023] | 60.80 | 31.06 | 55.80 | 58.80 | 37.10 | 48.71 | 45.69 |
| | C-TPT [CVPR 2024] | 61.2 | 25.6 | 54.8 | 59.7 | 35.7 | 47.4 | 44.0 |
| | TDA [CVPR 2024] | 61.35 | 30.29 | 55.54 | 62.58 | 38.12 | 49.58 | 46.63 |
| | BCA [CVPR 2025] | 61.81 | 30.35 | 56.58 | 62.89 | 38.08 | 49.94 | 46.98 |
| | **RTA** | **62.30** | **36.79** | **56.92** | **64.31** | **38.94** | **51.85** | **49.24** |
| | CLIP [ICML 2022] | 68.34 | 49.89 | 61.88 | 77.65 | 48.24 | 61.20 | 59.42 |
| ViT-B/16 | TPT [NeurIPS 2022] | 68.98 | 54.77 | 63.45 | 77.06 | 47.94 | 62.44 | 60.81 |
| | DiffTPT [ICCV 2023] | 70.30 | 55.68 | 65.10 | 75.00 | 46.80 | 62.28 | 60.52 |
| | C-TPT [CVPR 2024] | 69.3 | 52.9 | 63.4 | 78.0 | 48.5 | 62.4 | 60.7 |
| | TDA [CVPR 2024] | 69.51 | 60.11 | 64.67 | 80.24 | 50.54 | 65.01 | 63.89 |
| | MTA [CVPR 2024] | 70.08 | 58.06 | 64.24 | 78.33 | 49.61 | 64.06 | 62.56 |
| | Zero [NeurIPS 2024] | 70.89 | 64.03 | 65.11 | 80.82 | 50.32 | 66.24 | 65.03 |
| | Dyna [ICLR 2025] | 69.61 | 56.17 | 64.67 | 78.17 | 48.22 | 63.37 | 61.81 |
| | BCA [CVPR 2025] | 70.22 | 61.14 | 64.90 | 80.72 | 50.87 | 65.37 | 64.16 |
| | **RTA** | **71.13** | **65.65** | **65.43** | **81.05** | **51.23** | **66.90** | **65.84** |

**Single-label image classification** For the single-label classification on ImageNet and its variants datasets, we systematically evaluate the performance of two CLIP-based architecture models, `RN50` and `ViT-B/16`, as well as current mainstream TTA methods (as shown in Table 3). For the `RN50` architecture, RTA significantly outperforms existing methods on all evaluation metrics: it achieves $62.30\%$ accuracy on the ImageNet-1k dataset, an improvement of $2.49\%$ over the base CLIP model and $1.0 - 1.5\%$ higher than state-of-the-art methods such as TDA and BCA. RTA's advantages are even more pronounced on variant datasets with more pronounced distribution shifts, with an accuracy of $36.79\%$ on ImageNet-A, $5.73\%$ higher than DiffTPT, and an accuracy of $64.31\%$ on ImageNet-R, achieving a new best performance on this dataset.

Overall, RTA achieves an average accuracy of $51.85\%$ across the five datasets, and an OOD average accuracy (average of ImageNet-A, V2, R, and S) of $49.24\%$, improvements of $5.42\%$ and $6.15\%$, respectively, over the base CLIP model, validating its robustness to distribution-shifted scenarios.

For the `ViT-B/16` architecture, RTA also demonstrates leading performance: it surpasses methods such as Zero and BCA with an accuracy of 71.13% on ImageNet-1k, achieves an accuracy of 65.65% on ImageNet-A, an increase of 1.62% over the Zero method, and reaches an accuracy of 81.05% on ImageNet-R, further consolidating its ability to recognize complex variant data; its overall average accuracy reaches 66.90%, and the OOD average accuracy reaches 65.84%, which are 5.70% and 6.42% higher than the CLIP base model respectively.

Table 4: Accuracy comparison (%) on 10 cross-domain datasets for CLIP `RN50` and `ViT-B/16`.

| | Method | Pets | Flowers | Aircraft | DTD | EuroSAT | Cars | Food | SUN | Caltech | UCF | Average |
|---|---|---|---|---|---|---|---|---|---|---|---|---|
| **RN50** | CLIP [ICML 2022] | 82.97 | 62.77 | 16.11 | 40.37 | 25.79 | 55.89 | 74.82 | 60.85 | 87.26 | 59.48 | 56.63 |
| | TPT [NeurIPS 2022] | 84.49 | 62.69 | 17.58 | 40.84 | 28.33 | 58.46 | 74.88 | 61.46 | 87.02 | 60.82 | 57.66 |
| | DiffTPT [ICCV 2023] | 83.40 | 63.53 | 17.60 | 40.72 | 41.04 | _60.71_ | **79.21** | 62.72 | 86.89 | 62.67 | 59.85 |
| | C-TPT [CVPR 2024] | 84.0 | 65.3 | 17.5 | 43.1 | 29.4 | 57.3 | 76.0 | 62.1 | 87.4 | 60.7 | 58.3 |
| | TDA [CVPR 2024] | **86.18** | **68.74** | 17.61 | 43.74 | _42.11_ | 57.78 | 77.75 | 62.53 | 89.70 | 64.18 | 61.03 |
| | BCA [CVPR 2025] | 85.58 | _66.30_ | **19.89** | **48.58** | **42.12** | 58.13 | 77.19 | 63.38 | 89.70 | 63.51 | _61.44_ |
| | **RTA** | _86.08_ | 66.23 | _18.80_ | _46.58_ | 40.65 | **60.95** | _78.64_ | 63.58 | **89.94** | **65.42** | **61.78** |
| **ViT-B/16** | CLIP [ICML 2022] | 86.92 | 66.99 | 23.22 | 45.04 | 50.42 | 66.11 | 82.86 | 65.63 | 93.55 | 65.16 | 64.59 |
| | TPT [NeurIPS 2022] | 87.79 | 68.98 | 24.78 | 47.75 | 42.44 | 66.87 | 84.67 | 65.50 | 94.16 | 68.04 | 65.10 |
| | DiffTPT [ICCV 2023] | 88.22 | 70.10 | 25.60 | 47.00 | 43.13 | 67.01 | **87.23** | 65.74 | 92.49 | 62.67 | 65.47 |
| | C-TPT [CVPR 2024] | 87.4 | 69.9 | 23.9 | 46.8 | 48.7 | 66.7 | 84.5 | 66.0 | 94.1 | 66.7 | 65.5 |
| | TDA [CVPR 2024] | 88.63 | 71.42 | 23.91 | 47.40 | **58.00** | 67.28 | 86.14 | 67.62 | 94.24 | 70.66 | 67.53 |
| | MTA [CVPR 2024] | 88.24 | 68.06 | 25.20 | 45.90 | 45.36 | 68.47 | 85.00 | 66.67 | 94.21 | 68.69 | 65.58 |
| | Zero [NeurIPS 2024] | 87.20 | 66.82 | 24.42 | 45.86 | 43.77 | _68.48_ | 84.58 | 66.90 | 94.14 | 68.57 | 65.07 |
| | Dyna [ICLR 2025] | 88.28 | 69.95 | 24.33 | 47.96 | 42.28 | 67.65 | 85.42 | 66.32 | 94.32 | 68.72 | 65.52 |
| | BCA [CVPR 2025] | **90.43** | **73.12** | 28.59 | **53.49** | _56.63_ | 66.86 | 85.97 | **68.41** | 94.69 | 67.59 | _68.59_ |
| | **RTA** | _89.98_ | _71.80_ | **29.32** | _50.45_ | 53.65 | **70.40** | _86.45_ | _68.12_ | **95.80** | **70.98** | **68.70** |

**Cross-domain image classification** Table 4 presents the classification accuracy on 10 cross-domain datasets. Across 10 cross-domain datasets, RTA consistently outperforms prior adaptation methods for both CLIP RN50 and `ViT-B/16` backbones. For RN50, it raises the average accuracy to 61.78%, surpassing all baselines and yielding notable gains on domains such as Cars and UCF. For `ViT-B/16`, RTA achieves the highest average accuracy of 68.70%, edging out the previous best (BCA) while attaining top scores on fine-grained (Aircraft, Cars) and large-scale (Caltech) tasks. These results highlight RTA's strong generalization ability and its effectiveness in handling diverse and domain-shifted scenarios with both convolution-based and transformer-based CLIP models.

Table 5: Comparison of the mean average precision (%) of CLIP `RN50` on multi-label datasets.

| Method | MSCOCO | VOC2007 | NUSWIDE |
|---|---|---|---|
| CLIP [ICML 2022] | 47.53 | 75.91 | 41.53 |
| TPT [NeurIPS 2022] | 48.52 | 75.54 | 41.97 |
| DIffTPT [ICCV 2023] | 48.56 | 75.89 | 41.33 |
| DMN [CVPR 2024] | 47.53 | 75.91 | 41.53 |
| TDA [CVPR 2024] | 48.91 | 76.64 | 42.34 |
| RLCF [ICLR 2024] | 36.87 | 65.75 | 29.83 |
| ML-TTA [ICLR 2025] | _51.58_ | _78.62_ | _42.53_ |
| **RTA** | **53.25** | **80.20** | **45.52** |

Table 6: Comparison of the mean average precision (%) of CLIP `ViT-B/16` on multi-label datasets.

| Method | MSCOCO | VOC2007 | NUSWIDE |
|---|---|---|---|
| CLIP [ICML 2022] | 54.42 | 79.58 | 45.65 |
| TPT [NeurIPS 2022] | 53.32 | 77.54 | 46.15 |
| DIffTPT [ICCV 2023] | 53.91 | 77.93 | 46.13 |
| DMN [CVPR 2024] | 52.52 | 79.83 | 46.27 |
| TDA [CVPR 2024] | 55.21 | 80.12 | _46.72_ |
| RLCF [ICLR 2024] | 54.21 | 79.29 | 43.18 |
| ML-TTA [ICLR 2025] | _57.52_ | _81.28_ | 46.55 |
| **RTA** | **58.95** | **82.75** | **48.43** |

**Multi-label image classification** On the mainstream multi-label classification datasets MSCOCO, VOC2007, and NUSWIDE, we compare the mean average precision (mAP) performance of CLIP-based models with `RN50` and `ViT-B/16` architectures against state-of-the-art TTA methods (as shown in Table 5 and 6). For the `RN50` architecture, RTA achieves significant leading performance

across all datasets: on the MSCOCO dataset, it attains an mAP of 53.25%, surpassing the suboptimal method ML-TTA (51.58%) by 1.67% and outperforming the baseline CLIP model by 5.72%. On the VOC2007 dataset, it reaches an mAP of 80.20%, which is not only higher than TDA's 76.64% but also 1.58% higher than ML-TTA. On the NUSWIDE dataset with diverse categories, RTA achieves a new optimal result with an mAP of 45.52%, outperforming TDA and ML-TTA by 3.18% and 3.0% respectively, demonstrating strong adaptability to multi-label scenarios.

For the `ViT-B/16` architecture, RTA also maintains comprehensive advantages: on MSCOCO, it achieves an mAP of 58.95%, exceeding ML-TTA (57.52%) by 1.43% and the baseline CLIP by 4.53%. On VOC2007, it reaches an mAP of 82.75%, which is 2.63% higher than TDA (80.12%) and 1.47% higher than ML-TTA (81.28%). On NUSWIDE, it led all comparative methods with an mAP of 48.43%, outperforming TDA by 1.71% and ML-TTA by 1.88%.

Overall, RTA effectively improves multi-label classification performance across datasets for both model architectures, with more pronounced advantages in large-scale complex scenarios (e.g., MSCOCO and NUSWIDE), verifying its generalization effectiveness in general visual tasks.

## 5.2 FURTHER ANALYSIS

**Number of augmented views.** Figure 4 compares the classification accuracy of ImageNet (blue line) and ImageNet-variant (red line) at different numbers of augmented views. As the number of views increases, the accuracy of both datasets increases rapidly. Initially, the ImageNet-variant accelerates faster, even surpassing ImageNet at lower numbers of views. When the number of views exceeds 128, the accuracy stabilizes and the gap between the two approaches narrows, demonstrating that a sufficient number of views can reduce the impact of the dataset on model performance.

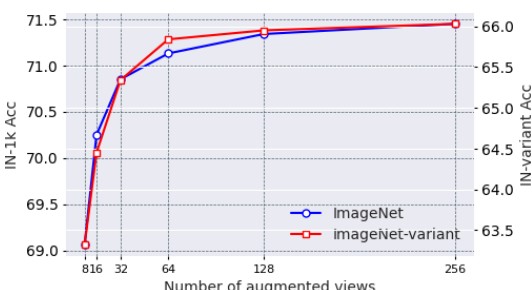

Figure 4: IN and its variants results on the different numbers of augmented views

**Number of regression mapping samples.** Figure 5 presents the classification accuracy of the ImageNet and ImageNet-variant datasets as the size of the sample pool (threshold $\geq 0.8$) increases. As the size increases, the accuracy of both datasets rises. In the early stage (when the size of the sample pool increases from $1,000$ to around $5,000$), the growth is significant, and the initial accuracy of ImageNet is slightly higher; as the size continues to increase (up to $50,000$), the accuracy of both continues to improve slowly and gradually stabilizes.

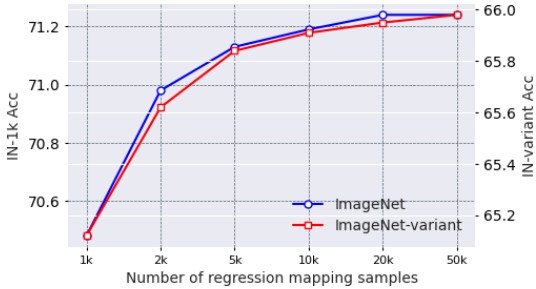

Figure 5: IN and its variants results on the different numbers of regression mapping samples

## 6 CONCLUSION

This paper proposes a Regression-based Test-Time Adaptation method (RTA) that aims at addressing the adaptation performance of vision-language models (such as CLIP) when there is a deviation between the test distribution and the pre-training data. Experiments reveal a strong regression mapping relationship between optimal augmented views and label cross-entropy loss, where a smaller loss indicates more accurate predictions from the corresponding view. Based on this, RTA pre-establishes the mapping relationship between views and cross-entropy loss by training a regression model on diversely distributed data. During testing, it can predict the loss of each augmented view without access to true labels and select confident views for integration. Experimental results show that RTA significantly outperforms existing entropy-based TTA methods on both single-label and multi-label datasets. Moreover, it only requires one training session to adapt to arbitrary test distributions, with negligible computational cost.

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

# Appendix for Regression-based Test-Time Adaptation of Vision-Language Models

## A  THE RISK OF DATA LEAKAGE

To address potential concerns regarding the use of the `ImageVal-12K` dataset and the possibility of data leakage, we provide detailed clarification and experimental results.

### A.1  DATASET USAGE

In our experiments, we utilize $1,000$ samples from the `ImageNet-12K` validation set solely for regression mapping. Tables 3, 4, 5, and 6 show that regression models trained on `ImageNet-12K` maintain stable and effective performance even under large distribution shifts. The entire data collection for the first stage is as follows:

**Step 1: Initial Filtering by Confidence.**  Starting from the `ImageNet-12K` validation set, we first remove any samples overlapping with the target dataset to avoid data leakage. From the remaining set, we select $5,000$ samples whose classification confidence scores are greater than or equal to $0.8$.

**Step 2: Logit Range Partitioning.**  For all selected samples, we calculate the maximum logit (`logit_max`) and minimum logit (`logit_min`). The logit range

$$[\texttt{logit\_min}, \texttt{logit\_max}]$$

is evenly partitioned into 10 intervals.

**Step 3: Stratified Sampling over Logit Intervals.**  From each interval, we randomly select 100 samples, resulting in a final regression training set of $1,000$ samples. This procedure ensures coverage of the entire logit value spectrum, enabling the regression model to learn a stable "*logit $\rightarrow$ loss*" mapping across varying magnitudes of logits.

### A.2  FURTHER EVALUATION OF GENERALIZATION

To more rigorously evaluate RTA's generalization ability, we additionally trained regression mappings using two alternative sources:

1. `ImageNet-1K` as the training samples
2. `ImageNet-V` as the training samples

In each case, the remaining datasets are used solely for the target sets.

Table 7: RTA generalization performance across different training and target sets. Results reported using CLIP-ViT-B/16.

| Training Set | ImageNet-I | ImageNet-A | ImageNet-R | ImageNet-K | ImageNet-V |
|---|---|---|---|---|---|
| CLIP baseline | 68.34 | 49.89 | 77.65 | 48.24 | 61.88 |
| RTA (`ImageNet-12K`) | 71.13 | **65.65** | **81.05** | 51.23 | **65.43** |
| RTA (`ImageNet-I`) | — | 65.34 | 81.02 | **51.76** | 65.38 |
| RTA (`ImageNet-V`) | **71.25** | 65.42 | 80.85 | 51.28 | — |

Even when the training and target datasets have different distributions, RTA demonstrates consistent and stable performance. During regression mapping, RTA learns a mapping from **logits** to **label cross-entropy losses**, which is inherently independent of the specific dataset distribution.

The only requirement for maintaining performance is the consistent use of the same vision-language backbone across both stages of the pipeline. For example, if CLIP-ViT-B/16 is used to obtain logits in the first stage, the TTA stage must also be performed on CLIP-ViT-B/16.

# B INFERENCE PROCESS OF RTA

This section clarifies how the RTA framework handles target datasets with arbitrary numbers of categories, without requiring re-training of the regression model.

**Stage 1: Regression Model Training.** The regression model, denoted as `Model_R`, is trained on the `ImageNet-12K` dataset, which contains $1,000$ categories serving as a fixed `base category set`. The training objective is as follows:

1. **Input:** Logits (similarity between image and class prompts) corresponding to the $1,000$ base categories. Each logit denotes a feature dimension.
2. **Output:** One scalar value, representing the predicted label cross-entropy loss.

**Stage 2: Adapting to Target Sets.** Given a target dataset with arbitrary novel categories (e.g., `Flower102` with 102 categories), the RTA inference stage proceeds as follows:

1. For each test sample, generate $n$ augmented views.
2. For each view, compute two sets of logits:
   - `logit_a`: $[n \times 1000]$ logits w.r.t. the fixed $1,000$ base categories, used solely for loss prediction via `Model_R`.
   - `logit_b`: $[n \times 102]$ logits w.r.t. the novel categories in the target dataset (`Flower102`), used for final classification output.
3. Feed `logit_a` into `Model_R` to predict the per-view loss:
$$\text{Pre}_{\text{loss}} \in \mathbb{R}^{n \times 1}$$
4. Rank the predictions in $\text{Pre}_{\text{loss}}$ and select the $top_k$ views with the lowest predicted losses as *confident views*.
5. Average `logit_b` across these confident views to produce the final prediction:
$$\text{Prediction} \in \mathbb{R}^{1 \times 102}$$

**Discussion.** In this design, the $1,000$ base categories from Stage 1 serve solely as a reference category set for learning the "*logit $\to$ loss*" mapping. This mapping is inherently decoupled from the specific label set of the target dataset. Consequently, for any dataset with arbitrary and novel categories, RTA uses the same base category logits to drive loss prediction, ensuring task independence and preserving generalization capability across variations in label cardinality.

# C INTEGRATING RTA INTO EXISTING TTA FRAMEWORKS

Table 8: Performance evaluation of integrating RTA into existing SOTA methods.

| Method | ImageNet-I | ImageNet-A | ImageNet-R | ImageNet-K | ImageNet-V |
|---|---|---|---|---|---|
| TPT | 68.98 | 54.77 | 77.06 | 47.94 | 54.77 |
| **TPT+RTA** | **70.21** | **58.43** | **79.32** | **49.05** | **55.98** |
| AWT | 71.32 | 60.33 | 80.64 | 51.60 | 65.15 |
| **AWT+RTA** | **72.45** | **66.86** | **82.35** | **51.95** | **66.21** |
| DPE | 71.91 | 59.63 | 80.40 | 52.26 | 59.63 |
| **DPE+RTA** | **72.34** | **66.32** | **82.05** | **53.65** | **65.96** |

The core principle of RTA is to replace the conventional entropy-based loss used for confident-view selection with a regression-predicted pseudo cross-entropy loss. This design enables RTA to be readily integrated into any TTA framework whose view-selection module is based on entropy loss, without requiring additional modifications.

We integrate our RTA into 3 TTA methods: TPT Shu et al. (2022), AWT Zhu et al. (2024b), and DPE Zhang et al. (2024), replacing the original entropy-loss selection module with RTA's loss predictor.

Across all datasets, integrating RTA consistently improves performance over the original methods. This demonstrates that RTA acts as an orthogonal enhancement, allowing it to serve as a *plug-and-play* module in diverse TTA frameworks.

## D  LOGIT VISUALIZATION AND CORRELATION ANALYSIS USING PCA

We employ PCA to visualize the correlation between logits and label cross-entropy loss as shown in Figure 6.

While the 2D PCA projection exhibits weaker color separation and clustering in compared to the t-SNE visualization in Figure 2 (due to inevitable information loss when nonlinear relationships are linearly projected), we conduct a quantitative correlation analysis between each principal component (PC) and the corresponding label cross-entropy loss in Table 9.

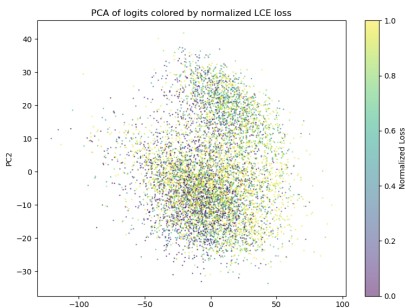

Figure 6: Logit visualization by PCA.

Results indicate that the first principal component is still significantly correlated with the loss (Pearson $r = 0.167$, $p \approx 9.1 \times 10^{-48}$; Spearman $\rho = 0.159$, $p \approx 1.0 \times 10^{-43}$), and several other principal components also demonstrate weaker yet statistically significant correlations. This confirms that even under purely linear projection, the relationship between logits and loss remains detectable.

Table 9: Correlation between principal components of logits (PCA) and label cross-entropy loss.

| PC Index | Pearson Corr | $p$-value | Spearman Corr | $p$-value |
|---|---|---|---|---|
| 1 | 0.167 | $9.1 \times 10^{-48}$ | 0.159 | $1.0 \times 10^{-43}$ |
| 2 | 0.025 | $3.2 \times 10^{-2}$ | 0.021 | $6.3 \times 10^{-2}$ |
| 3 | -0.054 | $3.5 \times 10^{-6}$ | -0.075 | $6.1 \times 10^{-11}$ |
| 4 | -0.087 | $3.8 \times 10^{-14}$ | -0.078 | $1.5 \times 10^{-11}$ |
| 5 | -0.002 | 0.87 | 0.008 | 0.46 |

However, the correlation strength observed in PCA components is clearly lower than that observed in the nonlinear t-SNE visualization, indicating that the mapping relationship predominantly exhibits a nonlinear structure. This finding validates our choice of a nonlinear regression model in RTA and motivates the selection of t-SNE, as it can more intuitively reveal the nonlinear structural characteristics between logits and label cross-entropy loss.

## E  IMPACT OF PSEUDO-LABEL QUALITY ON REGRESSION MODEL AND TTA PERFORMANCE

To evaluate the influence of pseudo-label quality on fitting the regression model and the resulting TTA performance, we conduct experiments on *ImageNet-A* in which ground-truth labels are progressively replaced with randomly assigned incorrect labels. The proportion of incorrect labels is varied from $0\%$ (true labels) to $100\%$ (all labels incorrect).

Table 10: Effect of pseudo-label noise levels on *ImageNet-A* accuracy (%). "True Label": regression model trained with ground truth. "Pseudo Label": regression model trained with CLIP pseudo-labels. $n\%$: $n\%$ pseudo-labels replaced with random incorrect labels.

| Model | Zero-shot | True Label | Pseudo Label | 20% | 40% | 60% | 80% | 100% |
|---|---|---|---|---|---|---|---|---|
| CLIP-ViT/B-16 | 49.89 | – | – | – | – | – | – | – |
| Shannon Entropy | 64.03 | – | – | – | – | – | – | – |
| **RTA** | – | **66.42** | 65.65 | 65.98 | 65.81 | 65.62 | 65.35 | 63.94 |

From Table 10, RTA achieves its highest accuracy (66.42%) when trained with true labels. As the noise proportion in pseudo-labels increases, performance gradually decreases; however, the decline is relatively small. Even in the 100% noise setting, RTA's accuracy (63.94%) remains only slightly

below that of Shannon entropy-based selection (64.03%), while substantially outperforming the zero-shot CLIP baseline (49.89%).

These results indicate that RTA is robust to moderate inaccuracies in pseudo-labels. In practice, pseudo-labels with approximately correct accuracy are sufficient to train a reliable regression model for confident-view selection during TTA.

## F  ANALYSIS OF ENTROPY VS. LABEL CROSS-ENTROPY UNDER DISTRIBUTION SHIFTS

### F.1  QUANTITATIVE COMPARISON IN ID AND OOD SCENARIOS

To investigate the deviation between Shannon entropy (HSE) and label cross-entropy (HLCE) under distribution shift, we compare their performance in all cross-domain out-of-distribution (OOD) scenarios.

Table 11: Accuracy (%) comparison for CLIP-ViT-B/16 with 64 views in cross-domain OOD datasets. HLCE consistently outperforms HSE across all domains.

| Method | Pets | Flowers | Aircraft | DTD | EuroSAT | Cars | Food | SUN | Caltech | UCF |
|---|---|---|---|---|---|---|---|---|---|---|
| CLIP-ViT-B/16 | 86.92 | 66.99 | 23.22 | 45.04 | 50.42 | 66.11 | 82.86 | 65.63 | 93.55 | 65.16 |
| $\mathbf{H}_{\text{SE}}(64)$ | 86.89 | 65.98 | 24.81 | 45.39 | 42.93 | 66.29 | 85.89 | 67.84 | 92.37 | 68.54 |
| $\mathbf{H}_{\text{LCE}}(64)$ | **94.99** | **80.05** | **52.87** | **63.12** | **71.48** | **90.42** | **96.65** | **88.42** | **98.99** | **87.44** |

In OOD scenarios—including the 10 cross-domain datasets, $\mathbf{H}_{\text{LCE}}$ yields substantially higher accuracy than $\mathbf{H}_{\text{SE}}$, often by more than 20 percentage points. These results validate that entropy is an unreliable confidence signal under strong distribution shifts, whereas $\mathbf{H}_{\text{LCE}}$ provides a more robust basis for confident view selection.

### F.2  VISUALIZATION OF VIEW-SELECTION DIFFERENCES

Here, we compare confident view selection based on $\mathbf{H}_{\text{SE}}$ and $\mathbf{H}_{\text{LCE}}$. For an image with 8 augmented views, we compute both logits and rank the views from lowest to highest loss.

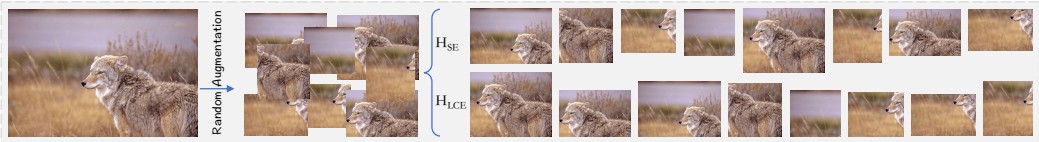

Figure 7: Comparison of confident view selection based on $\mathbf{H}_{\text{SE}}$ and $\mathbf{H}_{\text{LCE}}$

As shown in Figure 7, $\mathbf{H}_{\text{SE}}$ ranks one correct target-object view at the top, but places other views containing the target object lower in the ranking. In contrast, $\mathbf{H}_{\text{LCE}}$ consistently ranks nearly all semantically correct, high-confidence views at the top, leading to better semantic consistency and final prediction accuracy.

### F.3  DEFINITION OF CORRELATION COEFFICIENTS IN FIGURE 3

**Computation Process.**  Let $x_{i,j}$ denote the $j$-th feature dimension (i.e., class logit) of the $i$-th augmented view, and let $\ell_i$ be the corresponding LCE value computed as defined in Eq. (2). For each feature dimension $j$:

1. Collect $\{x_{i,j}\}_{i=1}^{N}$ across all $N$ augmented views in the regression-mapping dataset.

2. Collect the associated LCE values $\{\ell_i\}_{i=1}^{N}$ for the same views.

3. Compute the Spearman rank correlation coefficient $\rho_j$ between $\{x_{i,j}\}$ and $\{\ell_i\}$, which measures the strength of their monotonic relationship:

$$\rho_j = 1 - \frac{6 \sum_{i=1}^{N} d_i^2}{N(N^2 - 1)},$$

where $d_i$ denotes the difference between the ranks of $x_{i,j}$ and $\ell_i$.

4. Estimate the corresponding $p$-value $p_j$ to assess the statistical significance of $\rho_j$ under the null hypothesis of no monotonic correlation.

**Interpretation.** The resulting $(\rho_j, p_j)$ pairs capture both the strength and statistical significance of the relationship between a specific logit dimension and its associated LCE. High $|\rho_j|$ with low $p_j$ indicates a strong and statistically significant monotonic dependence, which guides the regression model in mapping logits to losses effectively.

## G  LOSS PREDICTION ACCURACY AND CROSS-DATASET GENERALIZATION OF REGRESSION MODEL

### G.1  LOSS PREDICTION ACCURACY AND CONSISTENCY

We assess the accuracy and trend consistency of the regression model's predicted cross-entropy loss versus the true label cross-entropy on `ImageNet-A`. The regression model is trained with pseudo cross-entropy derived from CLIP predictions.

For each augmented view, we obtain: (i) the predicted loss from the regression model and (ii) the ground-truth LCE computed according to Eq. (2) in the main text. These values are compared to evaluate both numerical deviations and ranking consistency.

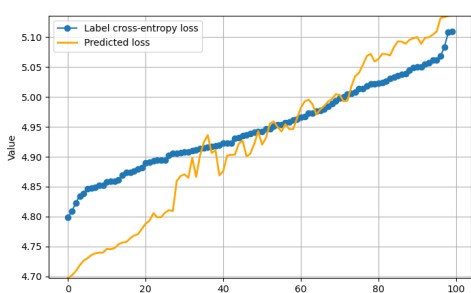

Figure 8: Logit visualization by PCA.

As shown in Figure 8, the absolute loss values differ numerically, but the overall trend between predicted and true losses is consistent. Since RTA's confident-view selection depends on *ranking* loss values rather than their absolute values, minor numerical differences do not affect selection accuracy as long as ranking trends are aligned.

### G.2  REGRESSION PERFORMANCE ACROSS ID AND OOD DATA

To further verify distributional generalization, we conduct an additional experiment: training the regression model on *ImageNet-I* or *ImageNet-V* and testing on the remaining datasets, ensuring that training and target distributions differ substantially.

Table 12: Accuracy (%) across various datasets when training the regression model on different sources. CLIP-ViT-B/16 backbone is used throughout. A slash "/" indicates that the dataset was used in training and is excluded from evaluation. Boldface denotes the best result in each column.

| Method | ImageNet-I | ImageNet-A | ImageNet-R | ImageNet-K | ImageNet-V |
|---|---|---|---|---|---|
| CLIP-ViT-B/16 | 68.34 | 49.89 | 77.65 | 48.24 | 61.88 |
| RTA (*ImageNet-12K*) | 71.13 | **65.65** | **81.05** | 51.23 | **65.43** |
| RTA (*ImageNet-I*) | / | 65.34 | 81.02 | **51.76** | 65.38 |
| RTA (*ImageNet-V*) | **71.25** | 65.42 | 80.85 | 51.28 | / |

The results indicate that RTA maintains stable accuracy even when training and target distributions differ significantly. This robustness is due to the regression stage learning a generalizable *logits →* *loss* mapping, which only requires logits extracted using the same vision–language backbone in both stages (e.g., CLIP-ViT-B/16).

The regression mapping learned on one dataset can be directly transferred to datasets from arbitrary distributions without retraining, ensuring that RTA can be deployed in diverse test-time adaptation scenarios.

## H CHOICE OF REGRESSION MODEL IN RTA

### H.1 EVALUATION OF NONLINEAR REGRESSION MODELS

To explore suitable regression models for predicting label cross-entropy from logits, we compare multiple models, including kernel-based methods, multilayer perceptrons (MLPs), and gradient-boosted decision trees.

Table 13: Accuracy (%) of various regression models integrated into RTA across different datasets. CLIP-ViT-B/16 backbone is used for logit extraction.

| Regressor | ImageNet-I | ImageNet-A | ImageNet-R | ImageNet-K | ImageNet-V |
|---|---|---|---|---|---|
| SVM (RBF kernel) | 70.34 | 64.89 | 80.54 | 51.03 | 65.05 |
| XGBoost | 70.65 | 65.12 | 80.87 | 50.98 | 65.34 |
| CatBoost | 71.02 | 65.45 | 80.89 | 51.05 | 65.25 |
| **LightGBM (RTA)** | **71.13** | **65.65** | **81.05** | **51.23** | **65.43** |

**Analysis.** Several nonlinear models achieve comparable accuracy; however, LightGBM offers the best balance across three criteria:

1. **Precision**: Highest or near-highest accuracy in all datasets tested.

2. **Robustness**: Strong handling of redundant or noisy features, which are common in high-dimensional logit vectors.

3. **Training Speed**: Requires only ∼5 seconds to train on 1,000 samples without complex feature preprocessing.

Tree-based models inherently capture nonlinear relationships and feature interactions, making them well-suited for modeling the logit-to-loss mapping.

**Conclusion.** While multiple nonlinear regression models are viable for RTA, LightGBM was selected for its superior combination of accuracy, robustness, and efficiency, ensuring the framework remains both effective and computationally lightweight in test-time adaptation scenarios.

## I ANALYSIS OF LOGIT–LOSS MAPPING.

We conduct analysis on both single-modal models (ResNet-50, ViT-B/16) and multi-modal models (CLIP-ResNet-50, CLIP-ViT-B/16). For each model, we extract the classification logits (single-modal) or image-text similarity logits (multi-modal) on ImageNet-1k and analyze their nonlinear correspondence with the ground-truth cross-entropy loss.

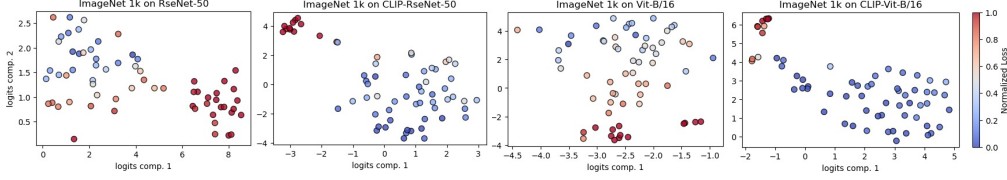

Figure 9: T-SNE between instance views' logits and label cross-entropy loss on single-modal models (ResNet-50, ViT-B/16) and multi-modal models (CLIP-ResNet-50, CLIP-ViT-B/16).

We find that all models exhibit a clear clustering structure in the logit–loss space, while the effect is substantially more pronounced in CLIP-based models. This suggests that the intrinsic structure of vision–language logits is particularly suitable for learning the logit-to-loss mapping through nonlinear regression.

## J STABILITY UNDER INPUT-SPACE PERTURBATIONS.

We evaluate how small perturbations in each model's *native input space* affect the resulting cross-entropy loss.

For single-modal CNN/ViT models (ResNet-50, ViT-B/16), we add a small pixel-level perturbation to the input image and measure the change in the final loss. For CLIP-based RTA, whose input is the logit vector, we directly perturb the logits and measure the corresponding change in label cross-entropy loss.

We ensure a fair comparison by matching perturbations in *relative magnitude*. Pixel values lie in $[0, 1]$ (average $\sim 0.5$), so a perturbation of $1/255$ corresponds to a relative scale of $0.4\%$–$0.8\%$. CLIP logits typically fall within $[10, 40]$ (average $\sim 25$), so an equivalent relative perturbation corresponds to $0.10$–$0.20$ in logit space.

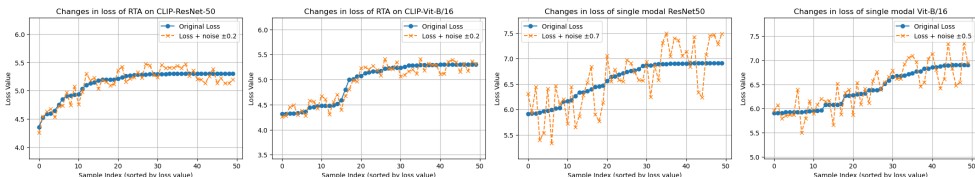

Figure 10: The loss change under perturbations on input space on single-modal models (ResNet-50, ViT-B/16) and multi-modal models (CLIP-ResNet-50, CLIP-ViT-B/16).

As summarized in Appendix Figure 10:

- **CNN/ViT models exhibit high forward sensitivity**: Even a $1/255$ pixel perturbation causes noticeable fluctuations in cross-entropy loss.

- **CLIP-based RTA shows substantially greater stability**: Under the same relative perturbation ($0.10$–$0.20$), the change in pseudo-LCE remains extremely small, indicating a smooth and structurally stable logit-to-loss mapping.

