# OpenReview forum: "Regression-based Test-Time Adaptation of Vision-Language Models"
_ICLR.cc/2026/Conference — Submitted to ICLR 2026_

### Official Review · Reviewer_Db4F · 2025-10-16

**Soundness:** 3
**Presentation:** 3
**Contribution:** 3
**Rating:** 8
**Confidence:** 2

**Summary:**

This paper aims to address the problem of how to select reliable augmented views for test-time adaption methods. Most of previous methods use the entropy of the distribution of a single test instance, while this paper proposes a new method by first training a regression decision tree for predicting the cross-entropy loss of each test instance and then selecting the one with minimal predicted cross-entropy loss. By integrating this new augmented view selection method and standard techniques for predictions, a new test-time adaption algorithm, RTA, is proposed.   The effectiveness of RTA is verified by extensive experiments.

**Strengths:**

* This paper is well-written and easy to follow.

* The technical contribution is significant. The proposed algorithm obtains better prediction performance than existing methods, and is much simpler and more efficient, as during training phase, training the regression model does not requires generate additional views on each test instance and during inference phase, the algorithm only needs to compute cross-entropy losses via the regress decision tree.

* The experimental results are sufficient.

**Weaknesses:**

* During the inference phase, the regression decision tree remains unchanged. Thus it may not adapt to the distributions of new test instances. Previous methods use memory modules to dynamically update the stored information and thus can adapt to non-stationary distributions.

* Some notations are confused. For instance, in Eq (6) and Eq (7), $\sum_{j:}$. The notation $j$ is unclear for me. In Eq (7), $y^{reg}_i$. The subscript $i$ is also unclear for me. In Eq (8) and line 313, the notation $x^{reg}_i$. I guess there are some typos on the subscripts.

**Questions:**

* The proposed method depends on the accuracy of pseudo-label. How do we guarantee the accuracy of those pseudo-label?


* Why is the decision tree selected? Could we use linear regression or non-linear regression model?

---

> ### Author Response · Authors · 2025-11-23
> **Official Response by Authors (1/2)**
>
> Thanks for your valuable suggestions, we will try to address your concerns and we are eager to engage in a more detailed discussion with you.
>
> `Weakness 1. Discuss Lack of Adaptation to Non-Stationary Distributions Due to Fixed Regression Decision Tree in Inference, Compared with Memory Module Approaches.`
> - Although the regression model remains fixed during inference, the regression stage in RTA essentially learns a **logits → loss** mapping, and the logits can originate from datasets of any distribution, as long as the same vision–language model is used in both stages (e.g., CLIP‑ViT‑B/16). As shown in Tables 3, 4 and 5, the regression model trained on ImageNet‑12K demonstrates stable performance gains across various target datasets, including ImageNet‑A/R/K/V, ten cross‑domain datasets (e.g., Pets, EuroSAT, Food‑101), and multi‑label datasets (MSCOCO, VOC2007, NUSWIDE).
>
> ||Imagenet-I|Imagenet-A|magenet-R|Imagenet-K|Imagenet-V|
> |-|-|-|-|-|-|
> |CLIP-ViT-B/16|68.34|49.89|77.65|48.24|61.88|
> |RTA (ImageNet-12k)|71.13|**65.65**|**81.05**|51.23|**65.43**|
> |RTA (ImageNet-I)|/|65.34|81.02|**51.76**|65.38|
> |RTA (ImageNet-V)|**71.25**|65.42|80.85|51.28|/|
>
> - To further evaluate RTA’s generalization, we separately use ImageNet-1K and ImageNet-V as regression training sets, with the remaining datasets serving as targets. Results show that even when the training set and target set have different distributions, RTA still maintains stable performance. In fact, during the regression mapping, RTA only learns the mapping from logits to label cross-entropy losses, which is independent of the specific dataset distribution.
> - Moreover, our method can be readily extended to an **online‑TTA** version, similar to memory‑based approaches: given that regression model training is extremely lightweight (e.g., ~5 seconds for 1,000 samples), we can store historical samples in a memory bank and periodically retrain the regression model either incrementally or fully. This can allow adaptation to non‑stationary distributions while maintaining high efficiency. We appreciate the reviewer for this insightful suggestion, which has inspired us to pursue future research on online regression mapping.
>
> `Weakness 2. Clarify Typos or Ambiguities in Eqs (6)–(8) and L313 (e.g., Meaning of j in Σ_j, i in y_i^{reg}, x_i^{reg}).`
>
> - In Eq. (6) and Eq. (7), the symbol $j$ denotes samples that fall into region $R_m$, i.e., the $m$-th leaf node of the decision tree.
> - In Eq. (7), the use of $i$ is a typographical error; the $i$ symbol should remove.
> - In Eq. (8), $i$ refers to the $i$-th augmented view.
> - We apologize for any confusion caused by unclear symbol definitions. In the revised manuscript, we have reviewed and corrected all instances of inconsistent notation throughout the paper.

---

> ### Author Response · Authors · 2025-11-23
> **Official Response by Authors (2/2)**
>
> `Question 1. Clarify How the Accuracy of Pseudo-Labels Is Ensured Given the Proposed Method’s Dependence on Them.`
> - We conduct a controlled experiment on the ImageNet‑A dataset to specifically evaluate RTA’s sensitivity to pseudo‑label accuracy:
>
> | **Model**         | **Zero-shot** | **True Label** | **Pseudo Label** | **20% Noise** | **40% Noise** | **60% Noise** | **80% Noise** | **100% Noise** |
> |-|-|-|-|-|-|-|-|-|
> | CLIP-ViT/B-16 | 49.89          | --             | --               | --            | --            | --            | --            | --             |
> | Shannon Entropy            | 64.03          | --             | --               | --            | --            | --            | --            | --             |
> | **ADTE**          | --            | **66.42**       | 65.65              | 65.98          | 65.81          | 65.62          | 65.35          | 63.54       |
>
> 1. Under the ideal condition of training with ground‑truth labels (true label cross‑entropy), RTA achieves highest accuracy of $66.42\\%$.
> 2. Gradually increase the noise ratio, randomly replace correct labels with incorrect ones, and observe the performance change.
> 3. As the incorrect‑label ratio increase, the performance decreases slightly, but even with $100\\%$ incorrect labels, RTA still significantly outperformed the CLIP baseline model and is only slightly lower than Shannon Entropy.
> - These results show that RTA is highly robust to pseudo‑label errors, and only requires reasonably accurate pseudo‑labels to train a reliable **logits → loss** regression mapping for robust confident‑view selection.
> - To further ensure pseudo‑label reliability, we can adopt strategies such as model ensembling, teacher‑student distillation, multi‑sample fusion, and raising the confidence threshold, thereby improving pseudo‑label accuracy and enhancing the regression model’s robustness.
>
> `Question 2. Explain Choice of Decision Tree as Regressor and Discuss Potential Use of Linear or Non-Linear Regression Models.`
>
> In fact, in the early of this paper, we have already evaluated both linear and nonlinear regression models for the **logits → pseudo‑label cross‑entropy loss** mapping, with the following conclusions:
>
> 1. Why not use a purely linear model:
> - We perform the PCA analysis again to analyze the correlation between the principal components and the ground‑truth label cross‑entropy loss.
>
> |PC Index|Pearson Corr|p-value|Spearman Corr|p-value|
> |-|-|-|-|-|
> |1|0.167|9.1E-48|0.159|1E-43|
> |2|0.025|0.032|0.021|0.063|
> |3|-0.054|0.0000035|-0.075|6.1E-11|
> |4|-0.087|3.8E-14|-0.078|1.5E-11|
> |5|-0.002|0.87|0.008|0.46|
>
> - From the result, although the first few principal components exhibit statistically significant correlations with the loss (e.g., PC1: Pearson = 0.167, p ≈ 9.1 × 10⁻⁴⁸; Spearman = 0.159, p ≈ 1.0 × 10⁻⁴³), the correlation strength is weak, and the loss distribution shows limited clustering in the $2D$ PCA projection (see appendix figure 6).
> - This indicates that the logits–loss relationship is predominantly **nonlinear**, making it difficult for linear regression to capture effectively. Figure 6 in the appendix further illustrates this pronounced nonlinear structure.
>
> 2. Why choose LightGBM/decision‑tree models:
>
> ||Imagenet-I|Imagenet-A|Imagenet-R|Imagenet-K|Imagenet-V|
> |-|-|-|-|-|-|
> |SVM|70.34|64.89|80.54|51.03|65.05|
> |Xgboost|70.65|65.12|80.87|50.98|65.34|
> |Catboost|71.02|65.45|80.89|51.05|65.25|
> |**Lightgbm (RTA)**|**71.13**|**65.65**|**81.05**|**51.23**|**65.43**|
>
> - We conduct a comprehensive evaluation of various nonlinear regression models, including kernel methods, MLPs, and gradient‑boosted decision trees. While several nonlinear models achieve comparable accuracy, LightGBM offered the best balance across **precision, robustness, and training speed** (~5 seconds for 1,000 samples).
> - Tree models inherently capture nonlinear relationships and feature interactions without complex feature preprocessing, and exhibit strong robustness to redundant features in the high‑dimensional logits space.

---

> > ### Comment · Reviewer_Db4F · 2025-11-26
> >
> > I thank the authors for detailed rebuttal that have addressed all of my concerns.
> >
> > I suggest that the authors include additional discussions on the adaption of the decision tree model,
> > rather than other non-linear regression models, in the revised manuscript. I will maintain the score.

---

> > > ### Author Response · Authors · 2025-11-26
> > > **Official Response by Authors**
> > >
> > > Thank you very much for your positive feedback and for taking the time to review our rebuttal. We appreciate your suggestion regarding a more detailed discussion on the choice of the decision tree model over other nonlinear regression models. We will incorporate this discussion into the revised manuscript to further clarify the rationale behind our design choices.

---

### Official Review · Reviewer_g4hE · 2025-10-24

**Soundness:** 2
**Presentation:** 3
**Contribution:** 2
**Rating:** 2
**Confidence:** 4

**Summary:**

This paper proposes a test-time adaptation (TTA) method for contrastive vision-language models, like CLIP.
The main approach of existing TTA methods for vision-language models is to select confident samples from augmented views of a test image.
The paper argues that prediction entropy, a measure used to quantify confidence, is not accurate for out-of-distribution data.
To accurately quantify the confidence for sample selection, the paper proposes training a regressor to predict cross-entropy loss and then utilizing the regressor for sample selection (regression-based test-time adaptation; RTA).
Experimental results demonstrate that RTA outperforms existing TTA methods on various datasets.

**Strengths:**

- Benchmark datasets are comprehensive in the experiment.

**Weaknesses:**

- W1: The key idea of the paper, training a loss predictor and selecting confident samples based on the predicted loss, is not novel [a], but [a] is not cited.
- W2: The problem of conventional entropy-based sample selection is not thoroughly examined.
  - It would be more convincing to investigate the deviation of entropy from label cross-entropy and validate that entropy is further unreliable in OOD.
  - It would be interesting to visualize cases where confident samples selected by entropy and cross-entropy differ.
  - The correlation analysis in Fig. 3 is hard to interpret. What is the "features" here? How were the correlation coefficients computed?
- W3: It is questionable whether the regression model is reliable since pseudo cross-entropy is employed for the labels of the training dataset for regressor $\mathcal{D}^\text{reg}$. This annotation process is similar to the sample selection process of existing methods.
- W4: It would be interesting to replace entropy with the regressor on the existing methods. Experimenting with this would demonstrate the applicability and generality of the proposed method.
- W5: Analysis of the trained regression model is limited. For instance,
  - How accurately can the regressor predict the loss? Does it align with the actual label cross-entropy while it is trained with pseudo cross-entropy?
  - Does the test accuracy further improve if the regressor is trained with label cross-entropy?
  - Evaluating regression performance on in-distribution and out-of-distribution datasets would be interesting.
- W6 (minor): L337 `we selecte ImageVal-12k`: typo?


[a] Kim et al., Learning Loss for Test-Time Augmentation, NeurIPS 2020.

**Questions:**

Please refer to the weaknesses.

---

> ### Author Response · Authors · 2025-11-23
> **Official Response by Authors (1/3)**
>
> Thanks for your valuable suggestions, we will try to address your concerns and we are eager to engage in a more detailed discussion with you.
>
> `Weakness 1. Clarify Novelty of Core Idea (Loss Predictor + Confident Sample Selection) and Add Missing Citation to [a].`
> - We thank the reviewer for pointing this out. RTA indeed involves the idea of “training a loss predictor and selecting samples based on the predicted loss.” However, this idea was primarily inspired by our investigation of **Test‑Time Linear Out‑of‑Distribution Detection** (TTLOD, CVPR 2024), rather than directly derived from **Kim et al (Learning Loss for Test‑Time Augmentation**, NeurIPS 2020). We will add a citation to Kim et al and clearly specify the similarities and differences:
> 1. **Motivation and scenario:** Our starting point is to generalize the concept in TTLOD (“performing OOD detection at test time via linear projection”) to the test‑time adaptation setting, and to select confident views under label‑free vision‑language models (e.g., CLIP). In contrast, Kim et al trained a loss predictor for supervised image classification tasks to perform test‑time augmentation, with a different scenario and objective.
> 2. **Training requirement:** Kim et al. require training within the target domain. In our case, the regression mapping is offline‑trained once using any domain pseudo‑labeled data and can be applied to any target domain without accessing ground‑truth labels or target‑domain samples.
> 3. **Model choice and use‑case:** We employ a tree‑based regression model to learn the nonlinear mapping between logits and label cross‑entropy loss offline, which only needs $5$ seconds of training cost. Kim et al used a deep neural network to predict loss, primarily in single‑view augmentation selection, and focused on improving image augmentation strategies during the test stage.
> - We have updated the Related Work section to include and discuss Kim et al, ensuring both the source of inspiration and our contributions are described accurately, and avoiding misunderstandings caused by missing related work.
>
> `Weakness 2.1. Investigate Deviation Between Entropy and Label Cross-Entropy, and Validate Entropy’s Unreliability in OOD Settings.`
> - We would like to clarify that Tables 1 and 2 already analyze the differences between Shannon Entropy (HSE) and label cross‑entropy (HLCE) in both in‑distribution (ID) and out‑of‑distribution (OOD) scenarios: ImageNet‑1K is ID, while IN‑A, IN‑V2, IN‑R, and IN‑K are OOD. The results show that in OOD scenarios, the differences between HSE and HLCE are pronounced.
>
> ||Pets|Flowers|Aircraft|DTD|EuroSAT|Cars|Food|SUN|Caltech|UCF|
> |-|-|-|-|-|-|-|-|-|-|-|
> |CLIP-ViT-B/16|86.92|66.99|23.22|45.04|50.42|66.11|82.86|65.63|93.55|65.16|
> |HSE(64)|86.89|65.98|24.81|45.39|42.93|66.29|85.89|67.84|92.37|68.54|
> |HLCE(64)|**94.99**|**80.05**|**52.87**|**63.12**|**71.48**|**90.42**|**96.65**|**88.42**|**98.99**|**87.44**|
> - In addition, we conduct further validation on the 10 cross‑domain datasets, as the above, all of which can be regarded as OOD scenarios that deviate more substantially from ImageNet. HLCE consistently delivers an overwhelming advantage over HSE.
>
> `Weakness 2.2. Visualization of Differences in Confident Sample Selection Between Entropy and Label Cross-Entropy.`
> - In Appendix Figure 7, we generate 8 augmented views for the image and compute HSE and HLCE. We then rank the views from lowest to highest loss. The results show that while HSE does select one view containing the target object at the top rank, other views that also contain the target object are ranked relatively low.
> - In contrast, HLCE is able to consistently rank almost all high‑confidence views containing the target object at the top. This demonstrates that HLCE‑based view selection is superior in terms of both semantic consistency and accuracy, and it explains the notable performance gains observed in our experiments.
>
> `Weakness 2.3. Clarify Fig. 3 Correlation Analysis: Definition of ‘Features’ and Method for Computing Correlation Coefficients.`
> - **Meaning of “features”:** Here, “features” refer to the per‑class logit values computed by the frozen image encoder and text encoder of CLIP. Each class‑specific logit is treated as one feature dimension.
> - **Computation of correlation coefficients:** Within the regression‑mapping dataset, for each feature dimension (i.e., class logit), we compute its Spearman rank correlation coefficient with the label cross‑entropy loss (LCE) of the view. Concretely, for a given feature dimension, we collect its values across all augmented views, and perform a correlation analysis with the corresponding LCE values (as defined in Eq. (2)), obtaining both the correlation coefficient and the *p*-value. These metrics reflect the strength and statistical significance of the monotonic relationship between a given feature and LCE.

---

> ### Author Response · Authors · 2025-11-23
> **Official Response by Authors (2/3)**
>
> `Weakness 3. Reliability of Regression Model Due to Use of Pseudo Cross-Entropy Labels, Which Resemble Existing Methods’ Sample Selection Strategies.`
>
> | **Model**         | **Zero-shot** | **True Label** | **Pseudo Label** | **20% Noise** | **40% Noise** | **60% Noise** | **80% Noise** | **100% Noise** |
> |-------------------|---------------|----------------|------------------|---------------|---------------|---------------|---------------|----------------|
> | CLIP-ViT/B-16 | 49.89          | --             | --               | --            | --            | --            | --            | --             |
> | Shannon Entropy            | 64.03          | --             | --               | --            | --            | --            | --            | --             |
> | **ADTE**          | --            | **66.42**       | 65.65              | 65.98          | 65.81          | 65.62          | 65.35          | 63.54       |
> - To evaluate the impact of pseudo‑label quality on the reliability of the regression model, we conducted a controlled experiment on the ImageNet‑A dataset:
> 1. Under the ideal condition of using ground‑truth labels, RTA achieved its highest 66.42% accuracy.
> 2. We then progressively increased label noise, randomly replacing a certain proportion of correct labels with incorrect labels, and observed the effect on TTA performance.
> 3. As the proportion of incorrect labels increased, RTA performance exhibited a slow downward trend, but the decline was relatively small: even with 100% incorrect labels, RTA’s performance was only slightly lower than that of Shannon entropy (HSE), and still significantly higher than the CLIP baseline model.
> - These results indicate that RTA is highly robust to pseudo‑label errors, and that low quality accurate pseudo‑labels are sufficient to train a reliable regression mapping model. We will include this noise‑sensitivity experiment and its discussion in the revised manuscript to clarify the reliability of the regression model.
>
> `Weakness 4. Demonstrate Applicability and Generality by Replacing Entropy with Proposed Regressor in Existing Methods.`
> - Our method is **orthogonal** to existing entropy‑based TTA approaches: as long as a method uses Shannon entropy (HSE) to rank/select confident views (e.g., TPT[1], DiffTPT[2], AWT[3], BoostAdapter[4], DPE[5]), we can directly replace its entropy‑based selection module with our regression‑based loss‑prediction model, without any other modifications. This replacement preserves the original framework while further improving the reliability of view selection.
>
> ||Imagenet-I|Imagenet-A|Imagenet-R|Imagenet-K|Imagenet-V|
> |-|-|-|-|-|-|
> |TPT|68.98|54.77|77.06|47.94|54.77|
> |**TPT+RTA**|**70.21**|**58.43**|**79.32**|**49.05**|**55.98**|
> |AWT|71.32|60.33|80.64|51.60|65.15|
> |**AWT+RTA**|**72.45**|**66.86**|**82.35**|**51.95**|**66.21**|
> |DPE|71.91|59.63|80.40|52.26|59.63|
> |**DPE+RTA**|**72.34**|**66.32**|**82.05**|**53.65**|**65.96**|
>
> - To validate this, we integrated our regression loss‑prediction model into TPT, AWT, and DPE by replacing their entropy‑ranking modules. The results (table above) show significant performance gains across multiple datasets, demonstrating the feasibility of our approach as a plug‑and‑play component within different TTA frameworks.
>
> [1]. Test-Time Prompt Tuning for Zero-Shot Generalization in Vision-Language Models. NeurIPS2022
>
> [2]. Diverse Data Augmentation with Diffusions for Effective Test-time Prompt Tuning. ICCV2023
>
> [3]. AWT: Transferring Vision-Language Models via Augmentation, Weighting, and Transportation. NeurIPS2024
>
> [4]. BoostAdapter: Improving Test-Time Adaptation via Regional Bootstrapping. NeurIPS2024
>
> [5]. Dual Prototype Evolving for Test-Time Generalization of Vision-Language Models. NeurIPS2024

---

> ### Author Response · Authors · 2025-11-23
> **Official Response by Authors (3/3)**
>
> `Weakness 5.1. Assess Loss Prediction Accuracy and Consistency with True Cross-Entropy Despite Pseudo Cross-Entropy Training.`
> - We conduct an experiment on the ImageNet‑A dataset to compare the cross‑entropy loss predicted by the regression model with the label cross‑entropy loss.
> - As shown in Appendix Figure 8, although there are some numerical differences between the two, their overall trends remain consistent.
> - Since the core objective of TTA is to rank loss values in order to select the most confident views, numerical differences in the losses do not affect confident‑view selection as long as the ranking trend of the losses is consistent.
>
> `Weakness 5.2. Investigate Whether Training Regressor with Label Cross-Entropy Improves Test Accuracy.`
> - As shown in the label‑noise sensitivity experiment described in `Weakness 3`, training the regression model with ground‑truth labels (true label cross‑entropy) achieves the highest performance, 66.42% accuracy on ImageNet‑A dataset, which is higher than when using pseudo‑labels. As the proportion of incorrect labels increases, accuracy gradually decreases; however, even with 100% incorrect labels, the performance remains higher than the CLIP baseline and only slightly below Shannon Entropy.
> - This indicates that while access to ground‑truth labels can yield further improvements, our method is highly robust to pseudo‑label noise, and reasonably accurate pseudo‑labels are sufficient to achieve near‑optimal performance.
>
> `Weakness 5.3. Assess Regression Performance on In-Distribution vs Out-of-Distribution Data.`
> - We would like to clarify that although the regression model’s training set is sourced from ImageNet‑12K, the model is applied during TTA to a wide variety of datasets with different distributions, including ImageNet‑A/R/K/V, $10$ cross‑domain datasets (e.g., Pets, EuroSAT, Food‑101), and multi‑label datasets (MSCOCO, VOC2007, NUSWIDE). As shown in Tables 3, 4, and 5, the regression model trained on ImageNet‑12K performs well across these OOD datasets.
>
> ||Imagenet-I|Imagenet-A|magenet-R|Imagenet-K|Imagenet-V|
> |-|-|-|-|-|-|
> |CLIP-ViT-B/16|68.34|49.89|77.65|48.24|61.88|
> |RTA (ImageNet-12k)|71.13|**65.65**|**81.05**|51.23|**65.43**|
> |RTA (ImageNet-I)|/|65.34|81.02|**51.76**|65.38|
> |RTA (ImageNet-V)|**71.25**|65.42|80.85|51.28|/|
>
> - To further verify the distributional generalization of the regression model, we added a new experiment: using ImageNet‑I or ImageNet-V directly as the regression training set and testing on the remaining datasets, i.e., where the training and target sets differ substantially in distribution. The results (specific values will be included in the final manuscript) indicate that RTA maintains stable performance even when the training and test distributions differ.
> - This is because the regression stage of RTA learns a **logits → loss** mapping, and the logits can be obtained from any dataset, as long as the same vision–language model is used in both stages (e.g., CLIP‑ViT‑B/16). This allows the learned mapping to be transferred to datasets from arbitrary distributions.
>
> `Weakness 6. Typo at L337 in ImageVal-12k Selection.`
> - We sincerely thank the reviewer for the careful and thorough review. In our experiments, we actually used the **validation set** of ImageNet‑12K and selected 1,000 samples from it as the regression training set. We will correct this description in the revised manuscript, and we will also carefully proofread the entire paper to check and fix any other potential typos or imprecise statements.

---

> > ### Comment · Reviewer_g4hE · 2025-11-25
> >
> > I appreciate the authors' response and additional experiments.
> >
> > My concerns about W2-W4 and W6 have been resolved.
> >
> > However, my most significant concern about novelty (W1) remains.
> > Although I understood the differences between RTA and Kim et al. in terms of inspiration source, motivation, and architectures, the rationale behind these differences and their significance should be clarified for novelty, as the method itself is similar.
> >
> > In fact, the method of Kim et al. can be applied to general classification models, including zero-shot classification with CLIP, and is competitive with RTA by replacing loss labels with pseudo-labels.
> >
> > Also, the `target data` in Kim et al. means the dataset on which the classification model was trained (Sec. 3.4).
> > So, RTA can be directly compared.
> >
> > For W5, directly evaluating the regressor's performance not only by resulting classification accuracy, e.g., by MSE between predicted and true losses, would be interesting.
> > As a result, it would be interesting to see if there are correlations between regression performance and downstream classification accuracy.

---

> ### Author Response · Authors · 2025-11-26
> **Official Response by Authors (1/2)**
>
> `Concern 1. Clarify Novelty with Kim et al., Learning Loss for Test-Time Augmentation, NeurIPS 2020.`
>
> - We acknowledge that the **regression‑mapping** idea in RTA shares conceptual similarities with the **loss predictor** proposed by Kim et al. While Kim et al. train a loss predictor using EfficientNet‑B0 for a single‑modality image classification task, our method employs a tree‑based regression model to learn the mapping from logits to cross‑entropy loss.
>
> - In fact, since CLIP was open source in $2021$, no prior work has examined that there exists a **cross‑dataset consistent and cross‑task stable nonlinear relationship** between CLIP **logits and label cross-entropy loss (LCE)**. RTA is directly motivated by our discovery of this structural phenomenon. In Section 4.1, we provide a systematic demonstration through t‑SNE analyses, Spearman correlations, and ceiling TTA performance, demonstrating that VLMs exhibit an **implicit logit structure** that does not depend on specific tasks or label spaces but emerges inherently from the **image–text matching paradigm**. Previous TTA methods have relied solely on per‑sample statistical information and have never leveraged such an inherent structural property. We believe this is a more valuable contribution beyond Kim et al., as we **establish and demonstrate it using both data analysis and controlled experiments**.
>
> - In addition, the loss predictor in Kim et al. involves a relatively complex design requiring multi‑layer features, additional regularization, a softmax function, a ranking‑based objective to optimize Spearman correlation, and an RNN to approximate the non‑differentiable Spearman metric. **In contrast, our research perspective is that we first observed that the logit-loss mapping exhibits a significantly clearer nonlinear structure. Therefore, we can directly compress these complex model structures, module designs, and loss optimizations into a simple regression model to directly learn the mapping from logit to loss, greatly reducing training and inference costs.**
>
> ||Imagenet-I|Imagenet-A|Imagenet-R|Imagenet-K|Imagenet-V|
> |-|-|-|-|-|-|
> |CLIP-ViT-B/16|	68.34|49.89|77.65|48.24|61.88|
> |Shannon Entropy|	70.89|64.03|80.82|50.32|65.11|
> |**RTA (lightgbm)**|	**71.13**|**65.65**|**81.05**|**51.23**|**65.43**|
> |RTA (deep model)|	70.25|62.35|78.48|50.09|63.89|
> - Here, we conduct a new experiment, using EfficientNet-B0 and other designed modules proposed by Kim et al. to replace the tree model in the regression mapping stage.
> - As can be seen, this structural nonlinear relationship is easier to learn with a dedicated machine learning model. The forward and backward propagation of the neural network will gradually amplify the changes in the values. The tree model is used to fit the mapping relationship from logit to loss, avoiding interference from other factors in the learning process.
>
> - Moreover, CLIP’s logits arise from the alignment between image and text features, rather than from a conventional supervised classifier’s linear head. Because text embeddings in CLIP are semantically structured (e.g., **cat’** and **tiger** occupy related structure in the embedding space), RTA can use the $1,000$ ImageNet‑12 classes as base categories to learn this structure and then generalize to arbitrary new classes and domains. By contrast, the loss predictor in Kim et al. is tied to EfficientNet‑B0’s single‑modality architecture, whose classification head encodes fixed class semantics and therefore cannot transfer to unseen categories.
>
> - Finally, RTA introduces a new paradigm for test‑time adaptation. Traditional TTA methods operate in an online manner: during inference, each sample’s prediction guides the selection or optimization of augmented views. In contrast, RTA learns a cross‑task **logit–loss mapping structure offline** using unlabeled data before testing, and applies this structure at test time at minimal cost. Thus, RTA is the first to shift TTA from **online per‑sample optimization** to **offline structural modeling**.
>
> - In summary, RTA contributes not only an efficient logit–loss regression mapping, but more importantly, reveals and exploits a previously unknown structural regularity within CLIP that is stable across tasks and categories. This insight enables us to move from online instance‑level TTA optimization to offline structural learning, thereby establishing a new paradigm for TTA research.

---

> ### Author Response · Authors · 2025-11-26
> **Official Response by Authors (2/2)**
>
> `Concern 2. Correlation between Regression Performance and Downstream Classification Accuracy.`
>
> - Thank you for the reviewer’s suggestion. We explore the correlation between regression performance and downstream accuracy from two perspectives. First, we adjust the **depth** and **number of leaves** in the tree model to explore regression performance. Second, we adjust the **number of regression mapping samples**. The results are shown below.
> (For RTA (x,y), x/y denotes the depth/number of leaves of the tree model. For RTA (N), N is the number of regression mapping samples.)
>
> | RTA (x,y) | train_mse | IN-I / mse | IN-A / mse | IN-R / mse | IN-K / mse | IN-V / mse |
> |-----------|-----------|------------|------------|------------|------------|------------|
> |RTA (3,4)     | 0.1253    | 70.78 / 0.1352 | 64.53 / 0.1396 | 80.68 / 0.1638 | 50.76 / 0.0975 | 64.64 / 0.1290 |
> |RTA (4,8)     | 0.1235    | 70.83 / 0.1259 | 65.39 / 0.1265 | 80.98 / 0.1496 | 50.95 / 0.0920 | 64.97 / 0.1172 |
> |RTA (5,16)    | 0.1179    | **71.13** / **0.1026** | 65.65 / 0.1145 | **81.05** / 0.1359 | **51.23** / 0.0827 | 65.43 / 0.1034 |
> |RTA (6,32)    | 0.0965    | 71.05 / 0.1042 | **65.86** / **0.1102** | 81.03 / **0.1317** | 51.06 / 0.0810 | **65.54** / 0.0985 |
> |RTA (7,64)    | 0.0928    | 70.98 / 0.1061 | 65.43 / 0.1109 | 80.97 / 0.1325 | 50.94 / **0.0805** | 65.12 / **0.0984** |
> |RTA (8,128)   | **0.0914**| 71.01 / 0.1076 | 65.48 / 0.1134 | 80.82 / 0.1328 | 50.98 / 0.0832 | 65.09 / 0.1003 |
>   - From the above results, as model complexity increases, the model's fitting ability gradually improves, thus the training MSE loss continuously decreases and reaches a stable state. Since each target dataset's distribution is different, there is no optimal depth or number of leaves that is optimal for all datasets.
>   - However, the accuracy of these target datasets all show a consistent trend of first increasing and then decreasing, while the MSE loss also shows a trend of first decreasing and then increasing. This means that lower-complexity tree models struggle to fit diverse target distributions, while higher-complexity models are prone to overfitting the training set, leading to reduced generalization ability.
>
> | RTA (N)   | train_mse | IN-I / mse | IN-A / mse | IN-R / mse | IN-K / mse | IN-V / mse |
> |-----------|-----------|------------|------------|------------|------------|------------|
> |RTA (100)     | **0.0371**| 70.65 / 0.1643 | 63.89 / 0.1432 | 79.65 / 0.1854 | 49.86 / 0.1154 | 63.76 / 0.1327 |
> |RTA (500)     | 0.0806    | 70.98 / 0.1287 | 65.03 / 0.1365 | 80.87 / 0.1487 | 50.76 / 0.1062 | 64.95 / 0.1238 |
> |RTA (1000)    | 0.0964    | 71.13 / 0.1066 | 65.65 / 0.1145 | 81.05 / 0.1359 | 51.23 / 0.0827 | 65.43 / 0.1034 |
> |RTA (5000)    | 0.0972    | 71.17/ 0.1056 | 65.89 / 0.1106 | 81.08 / 0.1345 | 51.31 / 0.0812 | 65.49 / 0.0994 |
> |RTA (10000)   | 0.0952    | 71.21 / 0.1034 | 65.98 / 0.1101 | 81.09 / **0.1332** | 51.32 / **0.0808** | **65.53** / **0.0925** |
> |RTA (50000)   | 0.0968    | **71.22** / **0.1032** | **66.01** / **0.1087** | **81.11** / 0.1333 | **51.35** / 0.0809 | **65.53** / 0.0954 |
>
> - From the above results, as the number of regression training samples increases, the training MSE loss gradually increases and then stabilizes. The MSE loss of the target dataset, on the other hand, gradually decreases and then stabilizes, while the accuracy gradually increases and then stabilizes.
> - This means that increasing the amount of regression training data can continuously improve the performance of the target dataset. However, since this training data is all from ImageNet-12k, a larger quantity cannot bring more diverse distributions, and the performance gradually tends to stabilize.

---

> > ### Comment · Reviewer_g4hE · 2025-11-27
> >
> > Thank you very much for the further explanations and experiments.
> >
> > I understood that CLIP has a special structure in the logit-loss space by learning image-text matching, which is a new paradigm compared to traditional classification models.
> >
> > I also recognized that RTA with a tree-based loss predictor outperformed the deep loss predictor models.
> >
> > However, the above response includes a new claim and hypotheses that should be verified.
> > For instance,
> > - `the logit-loss mapping exhibits a significantly clearer nonlinear structure`: Checking whether this phenomenon is specific to CLIP is necessary.
> > - `The forward and backward propagation of the neural network will gradually amplify the changes in the values.`: This should be verified by monitoring the learning process.
> > - Efficiency of the tree-based loss predictor: This is a new claim that was not presented in the paper.
> >
> > > RTA is the first to shift TTA from **online per‑sample optimization** to **offline structural modeling**
> >
> > I respectfully disagree, as Kim et al. and their subsequent works have already proposed training a loss predictor or augmentation function during or after pre-training phase before testing in TTA.
> >
> > To reflect the authors' new claims and discussion, the paper needs to be drastically reorganized.
> > I would like to keep my score because I think the paper is not yet ready for publication.

---

> ### Author Response · Authors · 2025-11-28
> **Official Response by Authors**
>
> `1、Regarding your question about whether the logit–loss mapping is a universal phenomenon.`
>
> - We conducted an additional analysis on both single‑modal models (ResNet‑50, ViT‑B/16)
> and multi‑modal models (CLIP‑ResNet‑50, CLIP‑ViT‑B/16).
> - Specifically, we extracted the classification logits from single‑modal models and the image–text similarity logits
> from CLIP models on ImageNet‑1k, and explored their nonlinear relationships with the
> corresponding label cross‑entropy loss. The visualizations are provided in Appendix Figure 9.
> - We observe that both single‑modal and multi‑modal models exhibit a logit–loss clustering pattern; however, the clustering is substantially more pronounced in the multi‑modal CLIP models, making them particularly well‑suited for learning the mapping through nonlinear regression.
>
> `2、Regarding your suggestion that “forward and backward propagation of the neural network should be verified.” `
>
> - For single‑modal CNN/ViT models, we applied small perturbations to the input images and monitored the change in the final label cross‑entropy loss.
> - For the logit‑based RTA mapping (used in CLIP models), we perturbed the logits directly and examined the change in the corresponding label cross‑entropy loss.
>
> Since these two strategies operate in different input spaces,  we apply perturbations in each model’s **native input space** to ensure a fair comparison:
>
> - **CNN forward path**:
>    -  image perturbation Δx → feature perturbation → logit perturbation Δl → loss change ΔL
> - **RTA forward path**:
>    -  logit perturbation Δl → regression mapping f(l) perturbation → loss change ΔL
>
> In CNNs, pixel values lie in $[0,1]$ with an average around $0.5$,  and the commonly used micro‑perturbation $\frac{1}{255}$ corresponds to a relative magnitude of approximately $0.4\\%–0.8\\%$.
> In CLIP‑based RTA, logits typically lie in the range $[10,40]$, with an average magnitude around $25$, so an equivalent relative‑scaled perturbation corresponds to $0.10–0.20$ in logit space.
> - This ensures that both models receive perturbations with comparable relative strength, avoiding unfair comparisons caused by differences in input scale.
>
> As shown in Appendix Figure 10, we obtain the following observations:
>
> - **CNN models show high sensitivity in their forward mappings**:
>   Even a $\frac{1}{255}$ pixel‑level perturbation produces noticeable fluctuations in the final cross‑entropy loss for ResNet‑50 and ViT‑B/16.
>
> - **The logit $\to$ loss mapping learned by RTA on CLIP‑ResNet‑50 and CLIP‑ViT‑B/16 is significantly more stable**:
>   Under the same relative perturbation scale (0.10–0.20), the corresponding changes in pseudo‑LCE remain extremely small, demonstrating a consistent level of smoothness and structural stability.
>
> `3、Clarify that RTA is the first to shift TTA from online per‑sample optimization to offline structural modeling.`
>
> - We would like to clarify that, in the context of **test-time adaptation (TTA)** for vision–language models such as CLIP, our work introduces an **offline structural modeling** paradigm that is capable of generalizing across datasets, label spaces, and domains. This shifts the TTA paradigm from the **conventional online per‑sample optimization** to a new form of **offline learning of a global logit–loss mapping**.
>
> - The methods we compare against are also **test-time adaptation (TTA)** methods designed specifically for vision–language models. Prior approaches all require online adaptation for each test sample. In contrast, RTA shows that, by exploiting the intrinsic logit–loss structure of CLIP, one can learn a **task‑agnostic, cross‑dataset mapping offline**, and then directly apply it at test time **without any online optimization or adaptation**.
>
> - This paradigm is fundamentally different from the **test-time augmentation (TTAug)** studied by Kim et al. Their work targets conventional single‑modal image classifiers and relies on predicting losses to choose augmentation operators within a single supervised task. It is essentially a form of **test-time data augmentation**, rather than test-time adaptation across tasks and domains for vision–language models.

---

### Official Review · Reviewer_ktWN · 2025-10-27

**Soundness:** 3
**Presentation:** 3
**Contribution:** 3
**Rating:** 6
**Confidence:** 3

**Summary:**

This paper investigates test-time adaptation (TTA) for adapting vision-language models (VLMs) like CLIP to downstream tasks. TTA for VLMs fundamentally involves quantifying uncertainty using entropy based on views generated by data augmentation, selecting samples with low uncertainty, and performing final predictions through their ensemble. Existing methods update partial parameters or learnable prompts by minimizing entropy, dynamically modifying the model during testing. The paper points out that the entropy used by these methods depends on the training distribution, limiting its effectiveness as a proxy metric for ground truth labels, particularly reducing its reliability for outliers. Instead of entropy, this paper measures view uncertainty by predicting cross-entropy loss against the ground truth label from VLMs' logits. Specifically, it first learns the mapping between logits and cross-entropy loss against the ground truth label using a nonlinear regression model like LightGBM by utilizing pseudo labels from CLIP. Next, during testing, the regression model predicts the cross-entropy loss for augmented views. These predictions are sorted in ascending order, and the subset with the lowest loss is used for prediction in an ensemble manner. Experiments confirm that the proposed method achieves competitive performance with existing TTA for VLMs without requiring parameter updates during testing.

**Strengths:**

- **S1.** The paper proposes an innovative TTA paradigm that predicts cross-entropy loss against correct labels from logits and utilizes this as an uncertainty metric. This holds promise as it may overcome the shortcomings of existing entropy minimization approaches.
- **S2.** The paper confirms the effectiveness of the proposed method across multiple tasks using a regression model trained solely on ImageNet-Val.

**Weaknesses:**

- **W1.** The preliminary experiments in Figure 2 were conducted using t-SNE, but t-SNE is not suited for accurately visualizing logits due to its inclusion of nonlinear transformations. Therefore, the claim regarding label loss clusters is somewhat misleading. The paper's argument would be further strengthened if the same discussion could be made using dimensionality reduction via linear transformations, such as PCA.
- **W2.** The paper does not discuss how the quality of pseudo labels generated by CLIP during training of the regression model on ImageNet-Val affects the performance of the regression model or TTA.
- **W3.** There remains a gap between the results using label loss in Tables 1 and 2 and the results of the proposed method in Table 3. The paper should discuss the cause of this gap and prospects for further reducing the performance gap.
- **W4.** The paper states in Section 1 L081 that “it can directly adapt to test instances with arbitrary distributions,” but the experiments in Section 5 cannot be considered exhaustive across domains. For example, accurate loss prediction is expected to be challenging on datasets from medical domains, such as microscope images or satellite images, which are not extensively represented in ImageNet.

**Questions:**

Please response the concerns raised in the weaknesses section.

---

> ### Author Response · Authors · 2025-11-23
> **Official Response by Authors (1/2)**
>
> Thanks for your valuable suggestions, we will try to address your concerns, and we are eager to engage in a more detailed discussion with you.
>
> `Weakness 1. Use Linear Dimensionality Reduction (e.g., PCA) Instead of t-SNE for Accurate Logit Visualization.`
> - On the same dataset, we conduct an analysis using PCA, and the visualization results are shown in Appendix Figure 6. Although the $2D$ PCA projection exhibits weaker color separation and clustering in the loss‑value distribution (an inevitable information lost when nonlinear relationships are linearly projected), quantitative correlation analysis (table below) shows that the first principal component is still significantly correlated with the loss (Pearson = $0.167$, $p ≈ 9.1 × 10^{-48}$; Spearman $= 0.159$, $p ≈ 1.0 × 10^{-43}$), and several other principal components also demonstrate weaker yet statistically significant correlations.
>
> |PC Index|Pearson Corr|p-value|Spearman Corr|p-value|
> |-|-|-|-|-|
> |1|0.167|9.1E-48|0.159|1E-43|
> |2|0.025|0.032|0.021|0.063|
> |3|-0.054|0.0000035|-0.075|6.1E-11|
> |4|-0.087|3.8E-14|-0.078|1.5E-11|
> |5|-0.002|0.87|0.008|0.46|
> - This indicates that even under purely linear projection, the relationship between logits and loss remains detectable, suggesting that the association is an intrinsic property of the logits rather than an artifact introduced by the nonlinear mapping of t‑SNE.
> - In addition, the correlation strength in the linear components is clearly lower compared to that observed in the nonlinear visualization (t‑SNE), implying that the mapping relationship predominantly exhibits a nonlinear structure.
> - This validates our choice of a nonlinear regression model. Therefore, in the paper, we select t‑SNE, as it can more intuitively reveal the nonlinear structural characteristics between logits and label cross‑entropy loss.
>
> `Weakness 2. Missing Discussion of CLIP Pseudo-Label Quality Impact on Regression Model and TTA Outcomes.`
> - To assess the impact of pseudo‑label quality on fitting the regression model, we use the ground‑truth labels as a baseline and progressively replace them with randomly assigned incorrect labels, observing the resulting effect on TTA performance on ImageNet‑A.
>
> | **Model**         | **Zero-shot** | **True Label** | **Pseudo Label** | **20% Noise** | **40% Noise** | **60% Noise** | **80% Noise** | **100% Noise** |
> |-------------------|---------------|----------------|------------------|---------------|---------------|---------------|---------------|----------------|
> | CLIP-ViT/B-16 | 49.89          | --             | --               | --            | --            | --            | --            | --             |
> | Shannon Entropy            | 64.03          | --             | --               | --            | --            | --            | --            | --             |
> | **ADTE**          | --            | **66.42**       | 65.65              | 65.98          | 65.81          | 65.62          | 65.35          | 63.94       |
>
> - From the above, RTA achieves the highest score of $66.42$ when using the true labels. As the proportion of incorrect labels increases, RTA’s performance gradually decreases; however, the drop is relatively small. Even when all labels are incorrect, RTA’s performance is only slightly below that of Shannon Entropy, while remaining significantly higher than the CLIP base model.
> - This result indicates that RTA is not particularly sensitive to pseudo‑label accuracy: pseudo‑labels with approximately correct accuracy are sufficient to train a reliable regression model.

---

> ### Author Response · Authors · 2025-11-23
> **Official Response by Authors (2/2)**
>
> `Weakness 3. Explain Causes of Performance Gap Between Label Loss Results (Tables 1–2) and Proposed Method (Table 3), and Discuss Ways to Further Reduce It.`
> - In Tables 1 and 2, $H_{LCE}$ refers to selecting the optimal view by directly computing the cross‑entropy loss for each view using the ground‑truth labels. This represents a supervised, information‑rich upper-bound scenario. In contrast, the RTA results in Table 3 do not access ground‑truth labels during actual inference. Instead, RTA relies on a pre‑learned regression mapping to predict the view losses under different distribution shifts.
> - Such predictions inevitably incur some error, leading to results lower than the ideal label‑loss upper bound. Moreover, the regression mapping is trained on a limited pseudo‑labeled dataset (1k high‑confidence samples, with no target‑domain samples) and can only approximate the relationship between LCE and logits. When the test and training distributions differ, the predicted losses may be biased.
> - Importantly, the relationship between logits and label loss can never be an exact identity mapping; it is inherently an *approximate nonlinear mapping* and therefore cannot be fully equivalent to the label loss.
>
> - **Possible future directions** include:
> 1. Increasing the quantity and diversity of training data for the regression mapping.
> 2. Employing more robust filtering strategies or semi‑supervised approaches to generate pseudo‑labels, reducing noise impact on regression fitting.
> 3. Expanding the feature set for the LightGBM model (via feature engineering or lightweight deep regression networks) to capture finer‑grained relationships between logits and label cross‑entropy loss.
> 4. Leveraging the speed of regression training (only ~$5$s for $1,000$ samples) to adaptively update the regression model during inference using historical samples, enabling online training and immediate target‑domain adaptation to mitigate distribution‑shift effects.
>
> `Weakness 4. Clarify Claim of Adapting to Arbitrary Distributions (L081) Given Limited Cross-Domain Experiments and Omission of Challenging Domains Such as Medical or Satellite Images.`
> - We sincerely thank the reviewer for the valuable feedback. We acknowledge that the experiments in Section 5 do not exhaustively cover all possible test domains, particularly extreme domains that differ substantially from the ImageNet distribution, such as medical microscopy images or remote‑sensing satellite imagery.
> - The statement in the introduction (line 081) — “can directly adapt to test instances from arbitrary distributions” — refers to the *domain‑agnostic* design philosophy of the RTA framework: the regression mapping only needs to be pre‑trained once on data with diverse distributions, and inference does not require domain‑specific online updates. It does *not* imply that we have already validated the method on all conceivable distributions.
> - We also acknowledge that in scenarios where low‑level visual features (e.g., spectral characteristics of satellite imagery or textural patterns in pathology slides) differ significantly from those in natural images, accurately predicting view losses becomes more challenging. In future work, we plan to incorporate greater diversity and extreme‑domain datasets into the regression training set to improve its coverage and further enhance RTA’s adaptability in a broader range of application scenarios.

---

> > ### Comment · Reviewer_ktWN · 2025-11-27
> >
> > Thank you for your detailed response.
> >
> > Regarding W1-3, the additional experiments and explanations have largely addressed my concerns. In particular, including these analyses and discussions in the main part would clarify the paper's argument. While it was somewhat disappointing that the verification across diverse distributions discussed in W4 was not shown with concrete results, I believe the paper can be improved by refining and clarifying the claims in the introduction and explicitly stating the discussion in the response as limitations in the main part.

---

> ### Author Response · Authors · 2025-11-27
> **Official Response by Authors**
>
> - We are very delighted to address part of the reviewer’s concerns, and we also recognize your strong interest in the generalization potential of RTA on extreme domains such as medical microscopy and satellite imagery. To this end, we apply RTA to the **origin CLIP**, **MedCLIP [1]** and **RemoteCLIP [2]** models, respectively, and evaluate them on $9$ domain datasets.
>
> - Taking MedClip as an example, in the first regression mapping stage, we use MedClip to extract logit and pseudo-label cross-entropy loss from ImageNet-12k, train the regression model $\mathbf{Model}_R$, and then use MedClip for the TTA process, observing the performance on the medical dataset. The results are as follows.
>
> |Model|RSI-CB256|WHU-earth|MLRSNet|RESISC45|RS2800|
> |-|-|-|-|-|-|
> |CLIP|37.35|51.18|55.29|60.92|59.31|
> |**CLIP (RTA)**|**39.42**|**53.81**|**57.13**|**62.26**|**60.11**|
> |RemoteCLIP|39.50|63.12|59.28|70.33|68.57|
> |**RemoteCLIP (RTA)**|**41.32**|**65.87**|**61.43**|**73.15**|**69.52**|
>
>
> |Model|CheXpert-5x200|MIMIC-5x200|COVID|RSNA|
> |-|-|-|-|-|
> |CLIP|0.2036|0.2254|0.5090|0.5055|
> |**CLIP (RTA)**|**0.2216**|**0.2365**|**0.5150**|**0.5231**|
> |MedCLIP|0.5942|0.5024	|0.7943|0.7682|
> |**MedCLIP (RTA)**|**0.6235**|**0.5421**|**0.8119**|**0.7753**|
>
> - These results show that even when there is a significant distribution gap between the regression training set and the target dataset, RTA still delivers stable performance improvements. This further supports our claim that the regression‑mapping stage of RTA primarily learns the **nonlinear relationship** between logits and loss, and therefore does not rely heavily on the specific source or distribution of the training data.
>
> - Of course, we also acknowledge that if the regression training set can include more general samples from extreme domains such as medical imaging or remote sensing, the performance of RTA in these domains could be further improved.
>
> - We hope these experiments can address the reviewer’s concerns, and we would be happy to further discuss any remaining questions.
>
> [1]. MediCLIP: Adapting CLIP for Few-shot Medical Image Anomaly Detection. (MICCAI 2024)
>
> [2]. RemoteCLIP🛰️: A Vision Language Foundation Model for Remote Sensing. (TGRS 2024)

---

### Official Review · Reviewer_HG6x · 2025-10-31

**Soundness:** 2
**Presentation:** 2
**Contribution:** 3
**Rating:** 6
**Confidence:** 3

**Summary:**

This paper proposes Regression-based Test-time Adaptation (RTA) for CLIP. Instead of selecting confident augmented views via entropy, the authors learn a regression mapping from view logits to cross-entropy loss using pseudo-labeled data gathered once from ImageVal-12k. At test time, RTA predicts a loss for each augmented view via the trained regressor, selects the top-k lowest loss views, and ensembles their logits for the final prediction. The paper motivates this by showing that if ground-truth cross-entropy is used for view selection (Ceiling TTA), accuracy improves significantly over entropy selection. Experiments on ImageNet variants, 10 cross-domain datasets, and multi-label benchmarks show that RTA outperforms state-of-the-art methods on RN50 and ViT-B/16 backbones.

**Strengths:**

1. This work treats confident-view selection as a supervised regression problem on logits, which is a new and interesting angle.
2. The authors motivated their approach clearly, by showcasing the Ceiling TTA ablation, which demonstrates large improvement over entropy.
3. Multiple evaluation settings are included, and the performance improvement is impressive on multi-label classification.
4. The paper reports nice ablations on number of augmented views and number of regression samples.
5. I thank the authors for the link to their code, however, I would appreciate it if they included a more complete README.

**Weaknesses:**

1. You mention that you use “ImageVal-12K” as the regression mapping data. I do not know this dataset, and I couldn’t find it by a quick search. Please add a reference to your text. Do you mean ImageNet-12K? Please elaborate on how close images in this dataset are to your target datasets. Isn’t there a chance of leakage?
2. Also, if the training data contains some of the “source” images, this can be seen as a deviation from standard TTA.
3. I believe it is not accurate to say that the mapping is independent of the downstream task. How do you handle different number of classes across datasets? I believe if the task changes, you need to retrain the regressor as the label set has changed. Please elaborate if my assumption is not correct.
4. I have concerns about the fairness of the comparison. While I understand your new take on the task, your approach uses additional samples and compute to train an auxiliary model while your competitors do not (please elaborate if they do). How do you argue the fairness of the comparison?
5. More discussion about your training data is needed. How are samples from your training dataset selected (besides the confidence threshold)? Do you explicitly try to include samples from a variety of classes, or is the selection completely random? A section in the appendix about your training data could help.
6. The mapping from baseline name to its reference should appear at least once somewhere in the paper. For example, TDA appears many times in text and tables, but it is not tied to Karmanov et al. This poses difficulties for a reader that is not an expert in the field. More importantly, I believe the references to some of these baselines might be missing from your references. For example, have you included the reference for Dyna?
7. Inconsistent venue for baselines across tables. For example:
	- TDA is listed as CVPR’24 and ICLR’25 in different parts of Table 4.
	- ML-TTA is lister as CVPR’25 and ICLR’25 in Tables 5 and 6.

My true score is closer to 5, but since only 4 or 6 are possible at this stage, I picked 6. I may update it after I read reviews from more confident reviewers.

**Questions:**

1. Your auxiliary model for picking augmented views is interesting and is applied on original CLIP. I am wondering about how orthogonal your idea is to SOTA methods. Have you tried using your regressor on top of SOTA methods to get further gains?
2. On L185, you claim near 100% accuracy on ImageNet-A, but this seems inconsistent with the results in Tables 1 and 2. Can you elaborate on this?
3. Performance gains are strong in Tables 3, 5, and 6, but small in Table 4 (cross-domain). Do you have an explanation on why your method is not as effective in this setting (compared to SOTA)?

---

> ### Author Response · Authors · 2025-11-23
> **Official Response by Authors (1/3)**
>
> Thanks for your valuable suggestions, we will try to address your concerns, and we are eager to engage in a more detailed discussion with you.
>
> `Weakness 1. Clarification and Citation for 'ImageVal-12K' Dataset and Potential Leakage Concerns.`
> - We appreciate the reviewer’s feedback and apologize for any confusion. In experiments, we use $1,000$ samples from the ImageNet-12K validation set for regression mapping.
> - To avoid data leakage, we remove all samples overlapping with downstream sets before regression mapping. Although ImageNet-12K and ImageNet-1K might be relatively close in distribution, the distribution gap is substantially larger with other targets (e.g., ImageNet-A/R/V/K, $10$ cross-domain datasets, MSCOCO, VOC2007, NUSWIDE). Results in Tables 3,4, and 5 show that regression models trained on ImageNet-12K maintain stable and effective performance even under large distribution shifts.
> - To further evaluate RTA’s generalization, we separately use ImageNet-1K and ImageNet-V as regression training sets, with the remaining datasets serving as targets.
>
> ||Imagenet-I|Imagenet-A|magenet-R|Imagenet-K|Imagenet-V|
> |-|-|-|-|-|-|
> |CLIP-ViT-B/16|68.34|49.89|77.65|48.24|61.88|
> |RTA (ImageNet-12k)|71.13|**65.65**|**81.05**|51.23|**65.43**|
> |RTA (ImageNet-I)|/|65.34|81.02|**51.76**|65.38|
> |RTA (ImageNet-V)|**71.25**|65.42|80.85|51.28|/|
>
> - Results show that even when the training set and target set have different distributions, RTA still maintains stable performance. In fact, during the regression mapping, RTA only learns the mapping from **logits** to **label cross-entropy losses**, which is independent of the specific dataset distribution.
> - The only requirement is that the same vision-language model (e.g., CLIP-ViT-B/16) be used consistently in two stages, i.e., if CLIP-ViT-B/16 is used in the first stage for obtaining logits, the TTA stage should also be performed on CLIP-ViT-B/16.
>
> `Weakness 2. Possible Deviation from Standard TTA If Training Data Includes Source Images.`
> - In Weakness $1$, we clarify that the regression training set contains no original images from the target datasets, avoiding data leakage. The experiments also show that RTA can still achieve stable performance improvements even when there exist significant distribution gaps between training and target.
> - This indicates that samples used in the regression stage can come from arbitrary sources and are not dependent on having consistent distributions with the target set.
> - For convenience and reproducibility, we chose the validation set of ImageNet‑12K as the regression training set.
>
> `Weakness 3. Questioning Task-Independence of Mapping: Need to Explain Handling of Different Class Counts and Potential Re-Training of Regressor.`
> - We appreciate the reviewer for pointing out the insufficient explanation of how RTA handles target datasets with an arbitrary number of labels.
> - In stage 1, the regression model (**Model_R**) is trained on ImageNet‑12K ($1,000$ categories as base):
> 1. **Input features:** The logits corresponding to these $1,000$ base categories (similarity scores between CLIP image embeddings and class prompts).
> 2. **Output:** A single scalar value, i.e., the predicted label cross‑entropy loss.
> - In stage 2, taking **Flower102** ($102$ novel categories) as the target set, the RTA process is as follows:
> 1. For each test sample, generate $64$ augmented views.
> 2. For each view, compute logits with respect to both the fixed $1,000$ base categories (from ImageNet‑12K) and the $102$ novel categories (from **Flower102**):
>
>    - a. **logit_a** = $[64 × 1000]$ — used for loss prediction by **Model_R**.
>    - b. **logit_b** = $[64 × 102]$ — used for final classification output.
>
> 3. Feed **logit_a** into **Model_R** to predict loss, **pre_loss** = $[64 × 1]$.
> 4. Rank **pre_loss** and select the $6$ views with the lowest losses as the **confident views**.
> 5. From **logit_b**, average the logits of these $6$ confident views to obtain the final output **prediction** = $[1 × 102]$.
> - Here, the $1,000$ categories from the first stage serve solely as a base category set, enabling the regression model to learn the “logit → loss” mapping. For any target dataset with arbitrary, novel categories, RTA uses the same base categories to generate logits for loss prediction, thereby achieving generalization across tasks and label cardinalities.
> - The key point is that the first stage of RTA learns only the mapping between logits and losses under a fixed vision–language model, and is independent of the label set in the target task. As long as the same vision–language model (e.g., CLIP‑ViT‑B/16) is used consistently in both stages, the learned mapping can be directly applied to datasets with any number of categories, without retraining the regression model.

---

> ### Author Response · Authors · 2025-11-23
> **Official Response by Authors (2/3)**
>
> `Weakness 4. Fairness of Comparison Given RTA’s Use of Extra Samples and Compute for Auxiliary Model.`
>
> **1. For Auxiliary Models**
> - **DiffTPT** [1]: Use Stable Diffusion to generate pseudo-images and construct labeled pairs for prompt tuning.
> - **RLCF** [2]: Employs larger teacher models (e.g., CLIP‑ViT‑L/14) to provide reward signals to a smaller student model (e.g., CLIP‑ViT‑B/16).
> - **AWT** [3]: Introduce LLMs (e.g., GPT) to produce diverse category descriptions.
> - **SwapPrompt** [4]: Maintains and distills parameters from historical models via EMA.
>
> In contrast, RTA trains a lightweight regression model in **5s** without requiring GPUs, and during inference simply replaces Shannon Entropy with a fast regression‑based prediction. No additional model cost is introduced.
>
> **2. For Auxiliary Data**
> - **TDA** [5]: Maintains positive/negative caches for subsequent predictions.
> - **DMN** [6]: Preserves historical pseudo-labels via a dynamic memory network.
> - **BCA** [7]: Uses historical samples for likelihood and prior adaptation.
>
> In contrast, RTA uses auxiliary data only once for **offline** regression mapping. After training, the model can operate independently without relying on continuous online data accumulation.
>
> **3. Summary**
> - While RTA introduces an additional regression training stage, its computation and storage costs are significantly lower than many existing TTA methods. It requires neither large‑model generation, nor distillation, nor online caching. Moreover, auxiliary data use is strictly **offline** and **one‑time only**, ensuring that the deployment‑phase inference setting remains unchanged.
>
> [1]. Diverse Data Augmentation with Diffusions for Effective Test-time Prompt Tuning. ICCV2023
>
> [2]. Test-Time Adaptation with CLIP Reward for Zero-Shot Generalization in Vision-Language Models. ICLR2024
>
> [3]. AWT: Transferring Vision-Language Models via Augmentation, Weighting, and Transportation. NeurIPS2024
>
> [4]. SwapPrompt: Test-Time Prompt Adaptation for Vision-Language Models. NeurIPS2023
>
> [5]. Efficient Test-Time Adaptation of Vision-Language Models. CVPR2024
>
> [6]. Dual Memory Networks: A Versatile Adaptation Approach for Vision-Language Models. CVPR2024
>
> [7]. Bayesian Test-Time Adaptation for Vision-Language Models. CVPR2025
>
> `Weakness 5. Clarification on Training Data Selection Criteria Beyond Confidence Threshold.`
> - Our training data is constructed as follows:
> 1. From the ImageNet‑12K validation set (after removing any samples overlapping with the target dataset), we select $5,000$ samples with classification confidence $≥ 0.8$.
> 2. For all selected samples, we obtain the maximum logit (**logit_max**) and minimum logit (**logit_min**), and divide the range **[logit_min, logit_max]** evenly into $10$ intervals.
> 3. From each interval, we randomly sample $100$ samples, resulting in a total of $1,000$ samples for the regression training set. This ensures that the dataset covers the entire logit value range, enabling the regression model to learn a stable “logit → loss” mapping for any size of logit.
> - We do not explicitly enforce class balance in sampling. The primary objective is to cover a diverse range of logit values rather than to represent specific categories.
>
> `Weakness 6. Ensure Baseline Names Are Mapped to References and Include Missing Citations (e.g., TDA, Dyna).`
> - We appreciate the reviewer’s reminder. We have revised the draft and explicitly cited each baseline method (e.g., TDA, Dyna, etc.) at least once so that readers can accurately match them to the corresponding references.
> - For Dyna (ICLR 2025, a method that dynamically samples the optimal prompt), we have included its performance comparison in Tables 3 and 4, but indeed missed the citation in the references. We have thoroughly checked all baseline citations and references to ensure their accuracy.
>
> `Weakness 7. Inconsistent Baseline Cite Labels Across Tables (e.g., TDA, ML-TTA).`
> - We have corrected the citation issue in the manuscript and have conducted a thorough check of all other references to ensure their accuracy.

---

> ### Author Response · Authors · 2025-11-23
> **Official Response by Authors (3/3)**
>
> `Question 1. Evaluate Orthogonality of Auxiliary View-Selection Model by Applying Regressor on Top of SOTA Methods.`
> - The core idea of RTA is to replace the original entropy-based loss with a regression-predicted pseudo cross‑entropy loss for selecting confident views. Therefore, as long as the view selection module of existing methods relies on entropy loss, RTA can be directly integrated without any other modifications. E.g., TPT [1], DiffTPT [2], AWT [3], BoostAdapter [4], and DPE [5].
>
> ||Imagenet-I|Imagenet-A|Imagenet-R|Imagenet-K|Imagenet-V|
> |-|-|-|-|-|-|
> |TPT|68.98|54.77|77.06|47.94|54.77|
> |**TPT+RTA**|**70.21**|**58.43**|**79.32**|**49.05**|**55.98**|
> |AWT|71.32|60.33|80.64|51.60|65.15|
> |**AWT+RTA**|**72.45**|**66.86**|**82.35**|**51.95**|**66.21**|
> |DPE|71.91|59.63|80.40|52.26|59.63|
> |**DPE+RTA**|**72.34**|**66.32**|**82.05**|**53.65**|**65.96**|
>
> - We replace the entropy‑loss selecting module with RTA’s regression predictor in TPT, AWT, and DPE. The above results show significant performance improvements across all datasets, demonstrating the feasibility of our approach as a plug‑and‑play component in diverse TTA frameworks.
>
> [1]. Test-Time Prompt Tuning for Zero-Shot Generalization in Vision-Language Models. NeurIPS2022
>
> [2]. Diverse Data Augmentation with Diffusions for Effective Test-time Prompt Tuning. ICCV2023
>
> [3]. AWT: Transferring Vision-Language Models via Augmentation, Weighting, and Transportation. NeurIPS2024
>
> [4]. BoostAdapter: Improving Test-Time Adaptation via Regional Bootstrapping. NeurIPS2024
>
> [5]. Dual Prototype Evolving for Test-Time Generalization of Vision-Language Models. NeurIPS2024
>
> `Question 2. Clarify Apparent Inconsistency Between L185 ImageNet-A Accuracy Claim and Tables 1–2 Results.`
> - In Table 2, using the CLIP‑ViT‑B/16 with $64$ augmented views, the accuracy on ImageNet‑A reaches $90.2\\%$, and the accuracy on ImageNet‑R reaches $94.4\\%$. Increasing the number of augmented views can further improve accuracy. This is why we stated that the performance was “close to $100\\%$.”
> - However, we acknowledge that this was not an accurate description, as the model’s performance had already approached a stable level. We have provided a more precise and accurate statement in the revised version.
>
> `Question 3. Explain Reduced Effectiveness of Method in Cross-Domain Setting (Table 4) Compared to SOTA.`
> - The cross‑domain benchmark itself is a challenging and diverse dataset, containing many fine‑grained domain categories. As shown in Table 4, no method achieves state‑of‑the‑art performance across all domains.
> - Our regression training set is sampled from ImageNet‑12K and does not include any cross‑domain categories, which may limit RTA’s performance on the cross‑domain benchmark.
> - Nevertheless, RTA achieves first or second place performance in most domains overall. If the regression training set also included some cross‑domain categories, RTA’s performance could be further improved.

---

> ### Comment · Reviewer_HG6x · 2025-11-27
>
> I appreciate the authors’ efforts in addressing my concerns. Their response resolves most of the issues I raised and provides many important details that were missing from the original submission. Based on this, I am willing to raise my score.

---

> ### Author Response · Authors · 2025-11-28
> **Official Response by Authors**
>
> We are very delighted to address your concerns, and we sincerely appreciate your thoughtful reassessment and raise the score, which has greatly improved the quality of our work. If you have any further concerns about this work, please feel free to contact us at any time.

---

### Public Comment · ~Yike_Yang1 · 2025-11-24
**Some questions regarding the method and experimental settings**

Thank you for sharing this interesting work. The idea of performing test-time view selection by learning a regression mapping between augmented views and their associated cross-entropy loss is quite appealing. The offline–online separation, as well as the motivation to overcome the limitations of entropy-based instance-level adaptation, is clearly presented. I appreciate the conceptual neatness of the proposed RTA approach.

I have a few questions and clarifications that I hope the authors can comment on.

1. About the notion of “free lunch”

The paper refers to the view–loss relationship as a “free lunch.” I am a bit unsure what “free” specifically refers to in this context. As I understand it, the method requires collecting an additional unlabeled dataset to train the regression model in advance. In many existing VLM-based TTA protocols (e.g., standard CLIP adaptation settings), methods typically adapt directly on the downstream test distribution without relying on external auxiliary data.

While the rebuttal mentions other methods such as TDA, BCA, and DMN as using auxiliary data, those approaches seem to rely mainly on historical test-time data rather than an external dataset.

2. On the number of regression-mapping samples

There seems to be a potential inconsistency regarding how many samples are used to train the regression model.

According to the implementation description and the rebuttal, the pipeline first obtains 5K pseudo-labeled samples using a confidence threshold of $0.8$, and then samples 100 examples per interval, resulting in 1K training samples. The experimental details also state that 1K samples are used for regression training.

However, in the subsection “Number of regression mapping samples” and in Figure 5, the text repeatedly refers to the x-axis values (1K, 2K, 5K, 10K, 20K, 50K) as “regression mapping samples” used “in the first stage.” For example, the paper states:

“Figure 5 presents the classification accuracy … as the number of regression mapping samples increases in the first stage.”

“As the number of regression mapping samples increases, the accuracy of both datasets rises.”

“In the early stage (when the number of samples increases from 1k to around 5k), the growth is significant …”

“… as the number of samples continues to increase (up to 50k) …”

These descriptions strongly suggest that the x-axis values reflect the actual number of samples used to train the regression model.

Yet, the largest pool described earlier in the paper consists of only 5K filtered samples, which makes it unclear how the ablation includes settings such as 20K or 50K. In addition, when comparing Figure 5 to the main results for CLIP ViT-B/16, the performance reported in the main experiments appears to align with the 5K setting rather than the claimed 1K training configuration.

For these reasons, the relationship between the sample counts in Figure 5 and the configuration used in the main experiments is not fully clear. I would appreciate clarification on how many samples are actually used for training in the main results, and how these choices correspond to the different sample sizes presented in Figure 5.

3. About the DPE results

In the rebuttal, you show that applying RTA within DPE yields a substantial improvement. This is an interesting result, and I wonder whether including the original DPE baseline and DPE+RTA in the main results table would help readers more directly appreciate the benefit of combining the two methods.

4. On using ImageNet logits as the base space for loss prediction across domains

I have another question regarding the applicability of the regression mapping when applying RTA to target domains whose label space does not overlap with ImageNet.

From the explanation in the rebuttal, the first-stage regressor is trained to map ImageNet-12K logits to predicted cross-entropy losses. For ImageNet and its variants, this seems reasonable because the target label space is still aligned with the original 1K ImageNet categories; in other words, for each test sample, one of the 1,000 logits corresponds to its true class, so the notion of “predicted loss” remains well-defined.

However, when applying RTA to datasets whose label spaces are completely disjoint from ImageNet-12K, none of the 1,000 ImageNet categories correspond to the true label of a test sample. In such cases, the cross-entropy loss computed against the 1K logits no longer has a ground-truth semantic meaning, and therefore it is unclear how the predicted loss from the regressor remains valid. Since the regressor is supposed to approximate the "ground-truth cross-entropy loss", I am having trouble understanding how this works when the base label set has no overlap with the target domain.

Could the authors clarify how RTA ensures that the predicted loss is meaningful when the target domain has no labels in common with ImageNet, and the “true class” for the cross-entropy does not exist in the 1K logits?

---

> ### Author Response · Authors · 2025-11-24
> **Official Response by Authors**
>
> We sincerely appreciate your recognition of our work and the thoughtful questions you have raised.
>
> `Question 1: Meaning of “free lunch”.`
>
> - As mentioned in our response to Reviewer g4hE (Weakness 1), our work is inspired by **Test Time Linear Out-of-Distribution Detection (TTLOOD, CVPR 2024)**, which learns a mapping between model predictions and OOD scores via linear regression.
> - In the abstract of **TTLOOD**, the authors explicitly refer to this paradigm as a **free lunch**. To avoid introducing new terminology that may cause potential ambiguity, we adopted the same term to describe the mapping from logits to label cross-entropy loss.
> - Here, **free** means that the regression mapping **is trained only once offline**, after which it can be directly applied at test time without any further training or updates. The computational cost of regression prediction is extremely low and essentially negligible. While many existing TTA methods also avoid training new models, they often require complex online algorithms or newly designed modules to adapt to each test sample. **RTA compresses these complex online adaptation processes into a single offline regression-mapping training stage, making the test-time adaptation process highly lightweight**.
>
> `Question 2: Number of samples for regression mapping.`
>
> - We apologize for the confusion in the paper. The values on the x-axis in Figure 5 actually represent the **candidate sample pool size**. As mentioned in our response to Reviewer HG6x (Weakness 5), we first obtain $5,000$ samples with confidence scores $>0.8$, then select $1,000$ samples at evenly spaced logit intervals for regression training. As the pool size grows, we proportionally increase the number of samples selected per interval.
>
> - Due to the highly non-uniform and long-tailed distribution of logits, **training set size** and **candidate pool size** increase together. For example:
>
>   - A training set of ~1k requires a candidate pool of ~5k;
>   - A training set of ~5k requires a candidate pool of ~20k;
>   - A training set of ~10k requires a candidate pool of ~30k.
>
> - This is because our **logit interval sampling strategy** requires a larger candidate pool to ensure uniform coverage over the logit range. The ImageNet-12K dataset is sufficiently large to support sampling of over 50k training instances without duplicates or sampling bias.
>
> `Question 3: Presentation of DPE+RTA results.`
>
> - We greatly appreciate your suggestion and agree that presenting the original DPE baseline alongside DPE+RTA results in the main results table would make the performance improvements more intuitive to readers.
> - We are revising the manuscript and will include this comparison in the main experimental table, along with supplementary “plug-and-play” results of RTA combined with other entropy-based methods.
>
> `Question 4: Applicability when the target label space does not overlap with ImageNet.`
>
> - In the first stage, the regressor learns the relationship between **logits (generated by a fixed vision–language model such as CLIP ViT-B/16)** and **label cross-entropy loss**. Crucially, during TTA, our goal is not to obtain an exact **absolute** loss value for an augmented view in the target domain (which may have no semantic meaning when label spaces do not overlap), but rather to obtain a **relative confidence ranking** between different augmented views.  As mentioned in our response to Reviewer g4hE (Weakness 5.1), although the label cross-entropy loss and predicted loss present numerical differences, their overall trends remain consistent.
>
> - For example, suppose the base categories in regression training are **{dog, cat, pig, sheep, cow}**, and the new target-domain category is **{tiger}**, which never appeared in training. **Tiger** is visually and semantically similar to **cat.** When computing similarity between a tiger image and the base categories, the logit for **cat** will be significantly higher than the others.
>
> - Now consider two augmented views:
>   - View A: Shows the full tiger body.
>   - View B: Shows only the tiger’s leg.
>
> - For these two views, the **cat** logit in View A would be higher than in View B. Even if the absolute loss value is imprecise, the **relative ranking** still holds—View A is ranked above View B due to higher confidence.
>
> - RTA relies precisely on this relative confidence ranking during TTA. The **logits–loss mapping patterns** learned during training generalize across domains because the similarity structure in the logits space continues to reflect semantic ordering, even when the category labels themselves are entirely disjoint.
>
> We hope the above clarifications address your concerns, and we welcome further discussion on any remaining questions.

---

> > ### Public Comment · ~Yike_Yang1 · 2025-11-24
> > **Thanks for Authors Response**
> >
> > Thank you very much for the detailed responses. I appreciate the clarifications, and I have a few follow-up comments that might help further improve the manuscript.
> >
> > 1. Clarification on the meaning of “candidate pool” and potential wording confusion
> >
> > Thank you for explaining that the x-axis in Figure 5 refers to the candidate pool size rather than the number of samples actually used for regression training. Your explanation makes sense.
> >
> > However, the current phrasing in the paper still reads very much like the x-axis represents the training sample size. This is also the common interpretation in other similar works, where “number of regression samples” typically refers to samples fed into the regressor rather than into the candidate pool. Therefore, it is quite natural for readers to assume that the values in Figure 5 denote the training data volume.
> >
> > To avoid this misunderstanding, I suggest adjusting the wording in the main text so that “candidate pool size” is explicitly distinguished from “training sample size.” This will make the methodology easier to follow for readers unfamiliar with your sampling strategy.
> >
> > Additionally, regarding the data source, I only found one publicly available version of ImageNet-12K, such as the HuggingFace dataset
> > https://huggingface.co/datasets/timm/imagenet-12k-wds/viewer/default/validation
> > ,
> > which contains roughly 65K validation images. Given that your rebuttal states:
> >
> > “From the ImageNet-12K validation set … we select samples with confidence ≥ 0.8,”
> > resulting in about 5K usable samples,
> >
> > it becomes unclear how a candidate pool of up to 50K samples can be formed after both (1) removing overlap with target datasets and (2) applying confidence filtering. This numerical relationship may require clarification in the final version.
> >
> > Additionally, could the authors clarify how duplicate samples are removed
> >
> > 2. Additional related work: PromptAlign (NeurIPS 2023)
> >
> > Since the rebuttal discusses auxiliary-data-based methods, I wanted to share one additional related work that may help contextualize your approach.
> >
> > PromptAlign (NeurIPS 2023) also utilizes an external dataset (ImageNet-1K, collecting 16 images per class to form a 16K auxiliary set). To my knowledge, it is the only other VLM-TTA method that explicitly introduces extra data besides the test stream.
> >
> > However, PromptAlign’s computational cost, adaptation procedure, and overall performance differ significantly from RTA. In fact, RTA appears much more efficient and achieves better accuracy while requiring only a single lightweight regression training stage. Adding a brief comparison or short discussion on PromptAlign could make your contributions even clearer, especially regarding efficiency and practicality.
> >
> > 3. Minor suggestions
> >
> > I hope the authors will consider releasing the regression-training data splits and code after acceptance, which would greatly facilitate reproducibility.
> >
> > Thank you again for the thorough responses. I hope to see this paper accepted.

---

> > > ### Author Response · Authors · 2025-11-25
> > > **Official Response by Authors**
> > >
> > > - Thank you very much for your suggestions on this work. We will follow your advice and upload a revised version as soon as possible.
> > > - In addition, regarding the ImageNet-12K dataset, the Hugging Face link you provided is in fact only a subset of ImageNet-12K. You can search for **“imagenet-12k-metadata”**, which corresponds to the complete ImageNet-12K dataset; the validation set contains 473k samples.
> > > - As for duplicate samples, ImageNet-1K is a subset of ImageNet-12K, so they can be directly removed by matching image filenames.
> > > - Once again, thank you for your attention to this work. We will organize the relevant code and data, and release them as open source after the paper is accepted.

---

### Comment · Area_Chair_UkP4 · 2025-11-25

Dear Reviewers,

Thank you for your time and effort in reviewing submissions for ICLR 2026. As we begin the author-reviewer discussion process, we kindly remind you to submit your responses to the author rebuttals by **December 2**.

Your engagement in this discussion phase is crucial to ensuring a fair and thorough evaluation of each submission.

### **Action Required**
- Carefully consider the authors’ rebuttal and any additional evidence they provide.
- Update your review (if applicable) to reflect your revised perspective.
- Discuss with the authors if further details are required

Your AC

---

### Meta-Review · Area_Chair_8qho · 2025-12-27

**Summary:**

This paper proposes Regression-based Test-Time Adaptation (RTA), which replaces entropy-based view selection with an offline-trained regressor that predicts cross-entropy loss from logits. The rebuttal resolved several methodological and presentation issues.
However, the recommendation is Reject due to remaining concerns about novelty, generalization, and the strength of the empirical evidence supporting the paper’s core claims. In particular, there is a substantial gap between the upper-bound scenarios using true label loss (Tables 1–2) and the performance achieved by the regression-based method, and the regression mapping, central to the contribution, shows limited and inconsistent gains in more challenging cross-domain settings. As a result, the evidence does not sufficiently justify claims of task- and distribution-independence of the regression model. Concerns regarding novelty relative to prior loss-prediction-based approaches also remain unresolved.

**Reviewer Concerns:**

The rebuttal clarified the regression training procedure, sampling strategy, and several technical details, and demonstrated that the method can be integrated into existing TTA frameworks. Additional analyses, including alternative visualizations and pseudo-label noise sensitivity, were helpful.
Nevertheless, key concerns remain unresolved. A major reviewer remained unconvinced about the novelty of the core idea, and some claims emphasized during the rebuttal would require stronger validation and substantial reorganization. The remaining gap to upper-bound performance and relatively weak cross-domain results raise doubts about the robustness and generality of the learned regression mapping. In addition, the use of the term “free lunch” appears overstated, given the need for offline regression training, unlike entropy-based TTA methods that require no offline training.

**Reviewer Scores:**

This submission received split reviews, with initial scores of 2, 6, 6, and 8. During the discussion, reviewer HG6x (initial score: 6) indicated willingness to raise their score, while reviewer g4hE (initial score: 2) stated that their assessment would remain unchanged. Given the initial average and that several key concerns remain partially unresolved after the rebuttal, I recommend Reject.

---

### Decision · Program_Chairs · 2026-01-26

Reject